# Clr4$^{SUV39H1}$ ubiquitination and non-coding RNA mediate transcriptional silencing of heterochromatin via Swi6 phase separation

Hyun-Soo Kim [1,2], Benjamin Roche[1,4], Sonali Bhattacharjee[1], Leila Todeschini [3], An-Yun Chang[1], Christopher Hammell [1], André Verdel [3] & Robert A. Martienssen [1,2] ✉

Transcriptional silencing by RNAi paradoxically relies on transcription, but how the transition from transcription to silencing is achieved has remained unclear. The Cryptic Loci Regulator complex (CLRC) in *Schizosaccharomyces pombe* is a cullin-ring E3 ligase required for silencing that is recruited by RNAi. We found that the E2 ubiquitin conjugating enzyme Ubc4 interacts with CLRC and mono-ubiquitinates the histone H3K9 methyltransferase Clr4$^{SUV39H1}$, promoting the transition from co-transcriptional gene silencing (H3K9me2) to transcriptional gene silencing (H3K9me3). Ubiquitination of Clr4 occurs in an intrinsically disordered region (Clr4$^{IDR}$), which undergoes liquid droplet formation in vitro, along with Swi6$^{HP1}$ the effector of transcriptional gene silencing. Our data suggests that phase separation is exquisitely sensitive to noncoding RNA (ncRNA) which promotes self-association of Clr4, chromatin association, and di-, but not tri- methylation instead. Ubc4-CLRC also targets the transcriptional co-activator Bdf2$^{BRD4}$, down-regulating centromeric transcription and small RNA (sRNA) production. The deubiquitinase Ubp3 counteracts both activities.

Heterochromatin has crucial roles in genome stability by silencing repetitive DNA and by suppressing recombination[1-7]. In the fission yeast *Schizosaccharomyces pombe*, the H3K9 methyltransferase Clr4, the homolog of mammalian SUV39H1 and SUVH39H2, methylates histone H3 lysine 9 (H3K9) to maintain centromeric heterochromatin during the cell cycle[8-10]. Centromeric non-coding RNA (ncRNA) is transcribed by RNA Pol II during S phase, and is processed to make centromeric sRNA by the concerted action of the RNAi transcriptional silencing complex (RITS), the RNA-directed RNA Polymerase Complex (RDRC) and Dicer (Dcr1)[7,11-16]. Di-methylation of H3K9 by Clr4, guided in part by sRNA, mediates co-transcriptional gene silencing (CTGS) of heterochromatin, which remains permissive to transcription during S phase. The subsequent tri-methylation of H3K9 by Clr4 and strong

binding of Heterochromatin Protein 1 (HP1) homologs Swi6 and Chp2, suppresses RNA Pol II-dependent ncRNA transcription via transcriptional gene silencing (TGS), which is essential for the epigenetic inheritance of heterochromatin[1,17-19]. However, the mechanism that promotes the transition from RNAi-dependent CTGS to TGS is unclear.

Ubiquitination can regulate enzyme activity by adding a single ubiquitin to its substrate (mono-ubiquitination), or target protein degradation via the 26S proteasome by the addition of a ubiquitin chain (poly-ubiquitination). Ubiquitination requires ubiquitin, an E1 ubiquitin activating enzyme, an E2 ubiquitin conjugating enzyme and an E3 ubiquitin ligase[20-23]. E2 enzymes interact with E1 and E3 enzymes and have essential roles in determination of ubiquitin linkage specificity and ubiquitin chain length[24]. The ubiquitination complex Cul4-

[1]Cold Spring Harbor Laboratory, Cold Spring Harbor, New York 11724, USA. [2]Howard Hughes Medical Institute, Cold Spring Harbor Laboratory, Cold Spring Harbor, New York, NY 11724, USA. [3]Institute for Advanced Biosciences, UMR InsermU1209/CNRS5309/UGA, University of Grenoble Alpes, Grenoble, France. [4]Present address: University of North Dakota, School of Medicine & Health Sciences, 1301 N Columbia Rd. Stop 9037, Grand Forks, ND 58202, USA. ✉e-mail: martiens@cshl.edu

Rik1^RaflRaf2 (CLRC) of *S. pombe* is composed of cullin Cul4, the RING-box protein Rbx1, the DNA damage binding protein 1 (DDB1) homolog Recombination in K 1 (Rik1) and two substrate receptors, Raf1 and Raf2, as well as the eponymous Clr4, which is only weakly associated in vitro[25–30]. Cul4 is a highly conserved E3 ligase and interacts with Rik1/Raf1/Raf2 for substrate recognition. All subunits of CLRC are essential for heterochromatin formation but its E2 enzyme and ubiquitination targets are not well characterized[25–28].

Liquid-liquid phase separation (LLPS) is the mechanism by which biomolecular condensates like cellular granules, including membraneless organelles such as the nucleolus, P granules, and stress granules form without a lipid bilayer. Multivalent macromolecular interactions, especially between proteins harboring intrinsically disordered regions (IDRs) and nucleic acids, drive formation of phase-separated liquid droplets. These biomolecular condensates have higher protein density and reduced internal diffusion, which contributes to a myriad of cellular functions, including heterochromatin formation[31–38]. Nucleosomes are post-translationally modified and many chromatin-related factors, such as heterochromatin protein 1 (HP1) and bromo-domain containing proteins (e.g. BRD4), which recognize these histone modifications, have an intrinsic propensity for LLPS and form phase-separated liquid droplets in vitro and in cells[39–43].

Here we set out to identify how the E3 ubiquitin ligase complex CLRC mediates the transition from RNAi-dependent CTGS to TGS. We found that Ubc4-CLRC mono-ubiquitinates Clr4 in the IDR, reducing its chromatin- and ncRNA-binding activities. Importantly, we found that Clr4 undergoes phase-separated liquid droplet formation along with Swi6 through its IDR. However, liquid droplet formation of Clr4 is blocked by the presence of ncRNA, which instead promotes self-association of Clr4, chromatin association of Clr4 and di-methylation of H3K9 but not tri-methylation of H3K9. We also found that the Ubc4-CLRC complex poly-ubiquitinates Bdf2 and inhibits the recruitment of Epe1 and Bdf2 to centromeric heterochromatin, reducing ncRNA transcription and sRNA production via RNAi. Both H3K9 tri-methylation by mono-ubiquitinated Clr4 and reduced sRNA production by targeted Epe1-Bdf2 are crucial for the transition between CTGS to TGS and spreading of heterochromatin, which can be reversed by overexpression of deubiquitinase Ubp3. These observations implicate phase separation as a critical step in the transition from RNAi to transcriptional gene silencing during heterochromatin formation.

## Results

### Ubc4 and Cul4 regulate heterochromatin silencing

We previously showed that a specific mutation in the E2 ubiquitin conjugating enzyme Ubc4 (*ubc4-G48D*) reduces heterochromatic silencing at centromeres, at the mating-type locus and at telomeres (Fig. 1a)[44]. Overexpression of Ubc4 also causes chromosome missegregation[45]. We found that the *ubc4-G48D* (*ubc4-1*) mutation results in Ubc4 protein instability leading to an overall lower Ubc4 protein level (Fig. 1b; Supplementary Fig. 1c). Replacing *ubc4* with its mouse homolog *UBE2D2* fully complemented *ubc4-1*, while *mUBE2D2-G48D* (*mUBE2D2-1*) had similar silencing defects to *ubc4-1*, showing that the function of Ubc4 is likely conserved throughout evolution (Fig. 1a; Supplementary Fig. 1a, b). We also isolated a spontaneous second-site suppressor S137T, which re-stabilizes Ubc4-1 protein levels, thus restoring silencing (Fig. 1b–d). The E3 ubiquitin ligase Cul4 is an essential subunit of the CLRC, and *cul4-GFP* (*cul4-1*) is a hypomorphic mutant with reduced heterochromatic silencing[27]. *ubc4-1* and the *cul4-1* mutants have very similar phenotypes, including loss of silencing of *ade6*⁺ and *ura4*⁺ reporter genes in centromeric heterochromatin, as

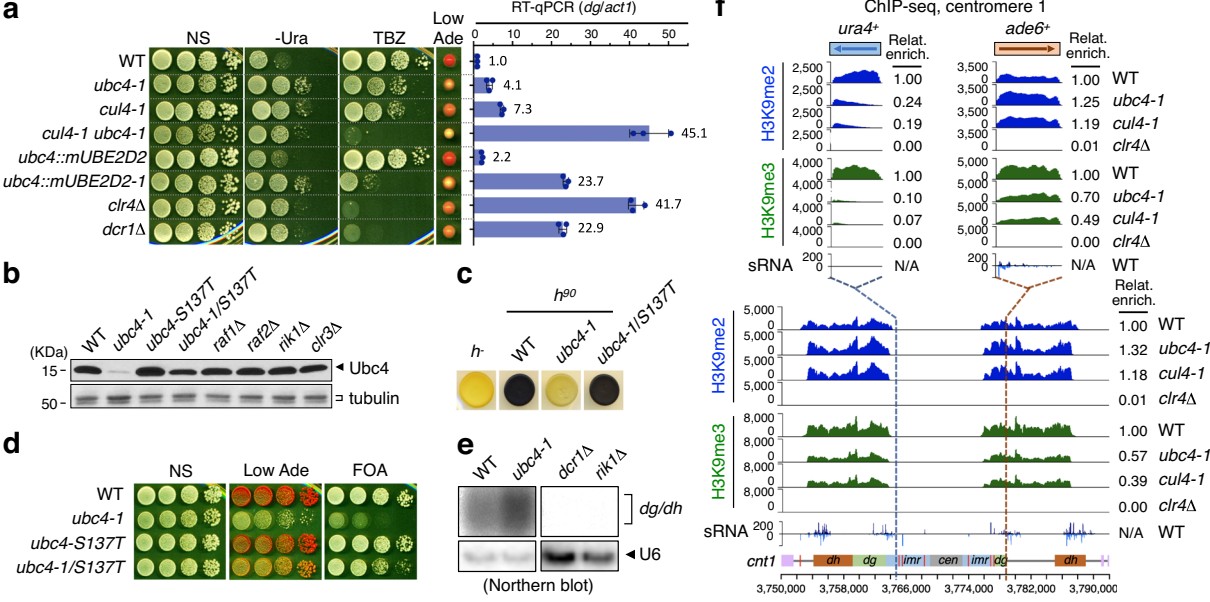

**Fig. 1 | Ubc4 and Cul4 regulate heterochromatin silencing. a** Growth spot assays for silencing of centromeric *ura4*⁺ and *ade6*⁺ reporter genes by failure to grow on no Uracil (-Ura) and colony color on low-adenine (Low Ade) media, and for chromosome segregation by growth on 15 μg/ml thiabendazole (TBZ) medium, were performed in the indicated strains. Colonies turn from white to red on low adenine media when *ade6*⁺ is silent. NS, non-selective medium. Right, RT-qPCR analysis of centromeric *dg* repeat (normalized to *act1*). Data are presented as mean ± SD (*n* = 3). **b** Ubc4 protein was greatly reduced in *ubc4-G48D* (*ubc4-1*) mutant cells. For WB analyses of Ubc4 protein in the indicated strains, antibodies against mUBE2D3 were used to detect Ubc4. WB of tubulin was used as loading control. Molecular weight markers are shown and uncropped images are provided in Source Data. **c** Mating-type switching defects in *ubc4-1* were suppressed by the secondary

mutation, *ubc4-S137T*. Non-switchable *h*⁻, WT and mutants in switchable strain *h*⁹⁰ were grown on sporulation medium and stained with iodine vapors to test switch efficiency and spore formation. Dark color (spore formation) indicates efficient switching and pale color indicates switch deficiency. **d** Assays for silencing of centromeric *ade6*⁺ and *ura4*⁺ on Low Ade and FOA media, a toxigenic substrate for Ura4 in the indicated strains. **e** *ubc4-1* mutants have increased centromeric sRNAs. sRNAs were isolated from indicated strains and analyzed by Northern blot with probes specific to *dg/dh* repeats and U6 for loading control. Uncropped images are provided in Source Data. **f** ChIP-seq reads of H3K9me2 and H3K9me3 mapped to chromosome 1 centromere in the indicated strains. ChIP-seq reads of H3K9me2 and H3K9me3 mapped to *ura4*⁺ and *ade6*⁺ reporter genes and sRNA-seq reads of WT are inset. N/A, not available.

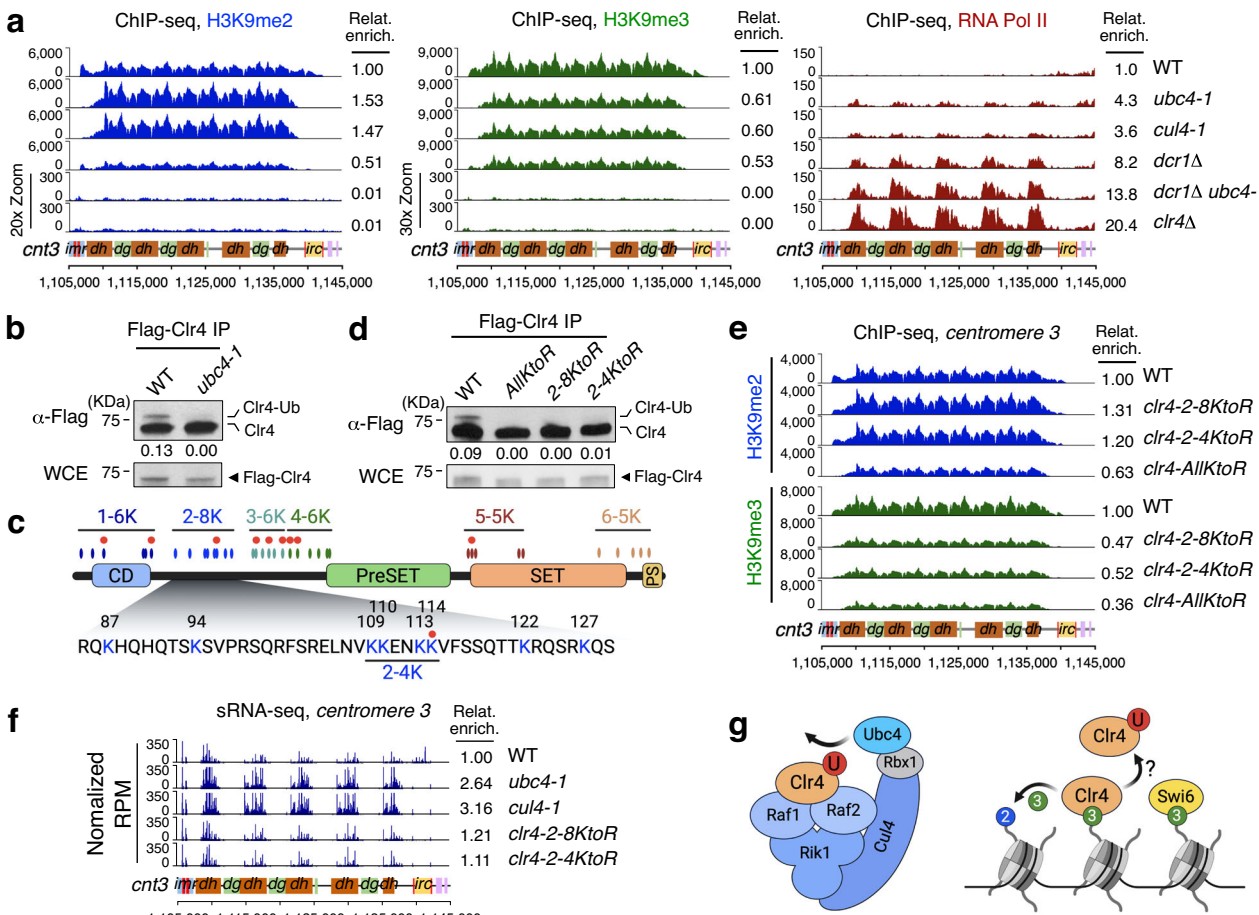

**Fig. 2 | Mono-ubiquitination of Clr4 by Ubc4-CLRC is required for H3K9me2 to H3K9me3 transition. a** Chromatin immunoprecipitation and sequencing (ChIP-seq) reads for H3K9me2 (blue), H3K9me3 (green) and RNA Pol II (Rpb1, red) were mapped to centromere 3 (*cnt3*) right arm in the indicated strains. Relative enrichments of ChIP-seq reads from each strain were divided by the total count in WT. Scales for *dcr1Δ ubc4-1* and *clr4Δ* H3K9me2 and H3K9me3 were expanded to visualize low reads. A schematic diagram of chromosome *centromere 3* right arm showing innermost repeat (*imr*), outermost repeats (*dg, dh*) and boundary region (*irc*) is shown. Vertical red lines and pink boxes represent tRNA and euchromatic genes respectively. **b**, **d** Western blot (WB) analyses of immunoprecipitated Flag-Clr4 (Flag-Clr4 IP) in the indicated strains. The ratio of ubiquitinated Clr4 (Clr4-Ub) compared to unmodified Clr4 is indicated below each lane. Bottom, WB analysis of Flag-Clr4 from whole cell extract (WCE). Molecular weight markers are shown and uncropped images are provided in Source Data. **c** Schematic diagram of domain

structure of Clr4 protein and the position of lysine residues. Ovals show the position of all 36 lysines of Clr4, which are arranged in 6 sub-groups. The second subgroup (2–8 K) is highlighted and amino acid sequences are shown. 2–4 K sub-group includes K109-K114. Red dots represent putative mono-ubiquitination sites identified by in vitro ubiquitination and Mass Spec analyses. CD: chromodomain, PS: PostSET. **e** ChIP-seq reads of H3K9me2 (blue) and H3K9me3 (green) mapped to chromosome 3 centromere right arm in the indicated strains. **f** sRNA-seq reads mapped to centromere 3 right arm in indicated strains. **g** Left, schematic diagram showing the presumptive subunit arrangement in the complex of Ubc4-CLRC and its substrate Clr4 for mono-ubiquitination. Red circle (U), ubiquitin. Right, model for the binding of Clr4 and Swi6 to existing H3K9me3 and the activity of Clr4 for converting H3K9me2 to H3K9me3. Blue circle, H3K9me2 and green circle, H3K9me3. Created in BioRender. Kim, H. (2024) BioRender.com/r28w184.

well as increased binding of Rpb1, the main subunit of RNA polymerase II, to heterochromatin (Figs. 1a, 2a) and increased production of siRNA (Figs. 1e, 2f, 5a). Interestingly, while H3K9me3 was reduced from both *ubc4-1* and *cul4-1* mutants at all heterochromatic loci tested, H3K9me2 was increased at the centromeric *dg* and *dh* repeats (Figs. 1f, 2a; Supplementary Fig. 2a). Increased H3K9me2 was also observed at the centromeric *ade6⁺* reporter gene, which accumulates small RNA, but not at the *ura4⁺* reporter gene, which does not (Fig. 1f; Supplementary Fig. 10a). The Swi6[HP1] protein, which coats and maintains heterochromatin, displayed intermediate binding levels at centromeres in *ubc4-1* and *cul4-1* mutants presumably because it recognizes both H3K9me2 and H3K9me3 (Supplementary Fig. 2b)[18,46]. In contrast, H3K9me2/3 and Swi6 were completely lost from both *ubc4-1* and *cul4-1* mutants at telomeres or mating-type loci where RNAi-dependent sRNA production is limited (Supplementary Fig. 2c, d).

To determine whether RNAi is important for maintaining H3K9me2 in *ubc4-1* mutant cells, we introduced the *ubc4-1* mutation in

*dcr1Δ* mutant cells and found that the *dcr1Δ ubc4-1* double-mutant lost both H3K9me2 and H3K9me3 and was hyper-sensitive to the microtubule destabilizing agent thiabendazole (TBZ), showing that centromeric heterochromatin was completely disrupted similarly to the *clr4* deletion mutant (Fig. 2a; Supplementary Fig. 2e). The more severe alleles *ubc4-3xHA* and *cul4Δ* abolished H3K9me2 and H3K9me3 altogether indicating that *ubc4-1* and *cul4-1* are hypomorphic and they might have other targets important for heterochromatin silencing (Supplementary Fig. 2f, g). In addition to their similar phenotypes, we found that Ubc4 and Cul4 physically interact and form a stable complex by co-immunoprecipitation (Co-IP) (Supplementary Fig. 3a). Ubc4 also forms a complex with Raf1 and Raf2, two substrate receptors of CLRC (Supplementary Fig. 3b) that mediate recruitment by siRNA-to-chromatin 1 (Stc1) and the RNA-induced transcriptional gene silencing (RITS) complex. The interaction between Ubc4-Cul4 and Ubc4-Raf2 was independent of nuclease treatment (Supplementary Fig. 3c, d), which indicates that Ubc4 and Cul4 directly form the E2/E3 complex by

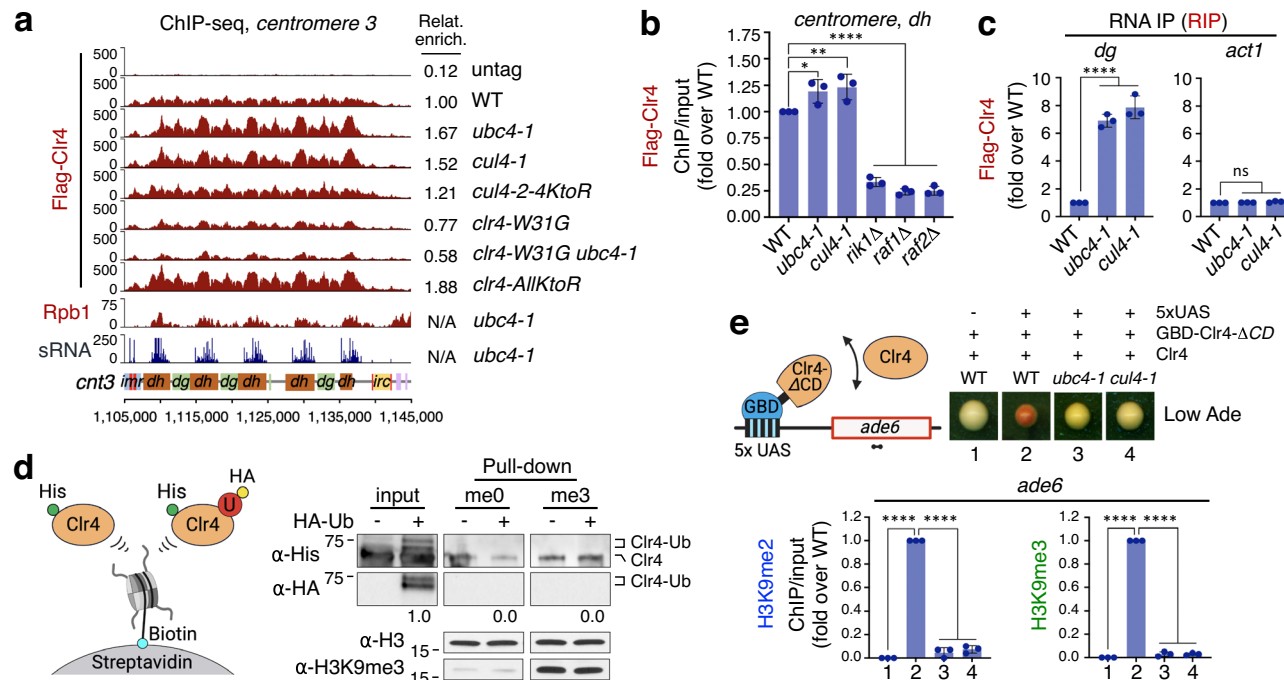

**Fig. 3 | Clr4 mono-ubiquitination induces its dissociation from chromatin and from centromeric ncRNA. a** ChIP-seq reads of Flag-Clr4 and RNA Pol II (Rpb1) and sRNA-seq reads mapped to chromosome 3 centromere (*cnt3*) right arm in indicated strains. N/A, not available. **b** ChIP-qPCR assay showing enrichment of Flag-Clr4 at centromeric *dh* repeat in the indicated strains. Data are presented as mean ± SD (*n* = 3). *P* values are from one-way ANOVA test (Dunnett's multiple comparisons test) without any adjustment. *P* = 0.0271 for *ubc4-1*, *P* = 0.0081 for *cul4-1* and *P* < 0.0001 for *rik1Δ*, *raf1Δ* and *raf2Δ*. **c** RNA immunoprecipitation (RNA IP) and qPCR of Flag-Clr4 to centromeric *dg* and *act1* transcript in the indicated strains. Data are presented as mean ± SD (*n* = 3). *P* values are from one-way ANOVA test (Dunnett's multiple comparisons test) without any adjustment. *P* < 0.0001 for *ubc4-1* and *cul4-1* (*dg*, left). ns, not significant (*act1*, right). **d** Binding assay of in vitro ubiquitinated His-Clr4 protein with nucleosomes with unmodified histone H3

(me0) or H3K9me3 histone (me3) anchored to streptavidin beads. Input and His-Clr4 proteins from pull-down were analyzed by WB analyses using antibodies as shown. U, ubiquitin. HA-Ub, HA-ubiquitin. Schematic diagram (left). Molecular weight markers (KDa) are shown on left side of panels (right) and uncropped images are provided in a Source Data file. **e** Schematic diagram for tethering of GBD-Clr4-*ΔCD* to 5xUAS-ade6 locus with free Clr4 protein. Primers for ChIP-qPCR are indicated (left). Right, assay for silencing of *ade6* on Low Ade medium in the indicated strains. Silencing (red color) depends on Ubc4 and Cul4. Bottom, ChIP-qPCR assays showing enrichment of H3K9me2 and H3K9me3 at *ade6* in the indicated strains. Data are presented as mean ± SD (n = 3). *P* values are from one-way ANOVA test (Dunnett's multiple comparisons test) without any adjustment. *P* < 0.0001 for 2, 3 and 4 (H3K9me2, Left). *P* < 0.0001 for 2, 3 and 4 (H3K9me3, Right). Created in BioRender. Kim, H. (2024) BioRender.com/r28w184.

## Mono-ubiquitination of Clr4 by Ubc4-CLRC is required for H3K9me2 to H3K9me3 transition

Next, we set out to identify the substrates of this novel E2/E3 complex, Ubc4-CLRC, for the regulation of H3K9 methylation. Interestingly, Clr4 co-purifies with the CLRC in vivo, but does not behave like a structural component of CLRC in vitro[25–28]. This led us to hypothesize that Clr4 is itself a target of the Ubc4-CLRC. Indeed, another Clr4 human homolog, SETDB1, is mono-ubiquitinated at its SET-Insertion domain and this modification is important for its H3K9 methyltransferase activity[47,48]. To test whether Clr4 is a substrate of Ubc4-CLRC, we purified 3xFlag-Clr4 protein from *S. pombe* cells and detected a single higher molecular weight form that reacted with ubiquitin antibodies in a Ubc4-dependent fashion (Fig. 2b; Supplementary Fig. 4a, b). A pulldown of ubiquitinated proteins using Ubiquitination Affinity Beads (UBA) revealed a similar higher molecular weight Clr4 which is also dependent on Ubc4 (Supplementary Fig. 4c). In addition to Ubc4, the presence of this higher molecular weight Clr4 depends on Cul4, Rik1, Raf1 and Raf2, the other components of CLRC, but not on the proteasome inhibitor MG-132 (Supplementary Fig. 4d, e). These results strongly suggest that Ubc4, together with CLRC, mono-ubiquitinates Clr4 in vivo.

We were unable to recover sufficient peptides in vivo for the identification of Clr4 ubiquitination sites by Mass Spectrometry because of the very low abundance of Clr4 in *S. pombe* cells[49], so

instead we ubiquitinated 6xHis-Clr4 recombinant protein in vitro. Clr4 was mono-ubiquitinated efficiently by UBE2D3, the human homolog of Ubc4, independently of an E3 ligase complex (Fig. 3d; Supplementary Fig. 5a–c), similarly to SETDB1[48,50]. This approach uncovered 9 possible ubiquitination sites, which mostly reside in the intrinsically disordered region (IDR) between the chromodomain (CD) and the PreSET domain (Fig. 2c and Supplementary Fig. 5a–d) which is involved in nucleosome binding[51].

To identify which of these represent functional sites in vivo, we partitioned the 36 lysines of Clr4 into 6 subgroups, which we then systematically mutated into arginines (R). We also included a mutant of all 36 lysines (*AllKtoR*). The all-lysine mutant (*AllKtoR*) and the second block mutant (*2-8KtoR*), corresponding to the middle of the IDR, completely abolished Clr4 ubiquitination in vivo (Fig. 2c, d; Supplementary Fig. 5d, e), consistent with the results of in vitro ubiquitination and Mass Spectrometry. By this means, we could narrow down the possible sites to 4 lysines (*clr4-2-4KtoR*; K109, K110, K113, and K114) (Fig. 2c, d; Supplementary Fig. 5f–i). These lysines are organized in tandem and are therefore very likely to compensate for each other. Using ChIP-seq, we found that *clr4-AllKtoR* lost both H3K9me2 and H3K9me3, presumably by reduced enzymatic activity[52]. *clr4-2-8KtoR* and *clr4-2-4KtoR* mutants had the same increase of H3K9me2 and decrease of H3K9me3 seen in *ubc4-1* and *cul4-1* hypomorphic mutants but loss of silencing of centromeric *ade6⁺* and *ura4⁺* reporter genes in *clr4-2-4KtoR* mutant was weaker than *clr4-AllKtoR* and *cul4-1* mutants (Fig. 2e; Supplementary Fig. 6a–c). In *ubc4-1* and *cul4-1* mutants, we

observed more than 2-fold increase of centromeric sRNA, as previously observed for *cul4-1*, but *clr4-2-8KtoR* and *clr4-2-4KtoR* mutants did not change centromeric sRNA production (Fig. 2f)[53]. These results indicate that Clr4 mono-ubiquitination by Ubc4-CLRC complex mediates the transition from H3K9me2 to H3K9me3 (Fig. 2g).

## Clr4 mono-ubiquitination induces its dissociation from chromatin and from centromeric ncRNA

Remarkably, the lysines in Clr4-2-8 co-incided precisely with two adjacent patches in the IDR that were previously shown to bind to the nucleosome core, independently of H3K9me3 (KKVFS and KRQSRK)[51]. Consistently, Clr4 binding to chromatin was increased in *ubc4-1* and *cul4-1* mutants, especially at *dh* centromeric repeats and at the *ade6⁺* reporter gene, where sRNAs are over-produced (Fig. 3a; Supplementary Figs. 7a, b, 10a). This increased binding of Clr4 to chromatin is not dependent on its chromodomain because there was no change in combination with a chromodomain mutant (*clr4-W31G ubc4-1*, Fig. 3a; Supplementary Fig. 7a)[18,53]. Note that the *clr4-2-4KtoR* mutant has much weaker binding than *ubc4-1* or *cul4-1* mutants presumably because *clr4-2-4KtoR* mutation does not affect ncRNA transcription and sRNA production (Figs. 2f, 3a; Supplementary Fig. 6c). Mutating all lysines (*AllKtoR*) resulted in an even stronger affinity of Clr4 to chromatin (Fig. 3a; Supplementary Fig. 7a). In contrast, Flag-Clr4 binding to centromeric heterochromatin decreased when H3K9me2/3 was completely abolished by *rik1Δ*, *raf1Δ* and *raf2Δ* (Fig. 3b)[26]. Clr4 also binds centromeric ncRNA[54] and this activity was strongly increased in *ubc4-1* and *cul4-1* mutants and slightly but significantly increased in the *clr4-2-4KtoR* mutant (Fig. 3c; Supplementary Fig. 6d). These results indicate that both chromatin and ncRNA binding activities of Clr4 are down regulated by Clr4 mono-ubiquitination.

To further explore the effect of Clr4 mono-ubiquitination on chromatin binding, we carried out nucleosome binding assays using in vitro ubiquitinated 6xHis-Clr4 recombinant protein and 3xFlag-Clr4 purified from *S. pombe* cells, with nucleosomes with either unmodified histone H3 (me0) or H3K9me3 (me3) tethered to streptavidin beads. In contrast to unmodified Clr4 protein, which had strong binding affinity to the nucleosome, ubiquitinated Clr4 did not bind to nucleosomes (Fig. 3d; Supplementary Fig. 7c). These results suggest that Clr4 mono-ubiquitination promotes dissociation of Clr4 from chromatin in agreement with the in vivo Clr4 ChIP-seq data.

Next, we tested whether Ubc4-Cul4 ubiquitination activities are important in vivo in the context of a minimal silencing system, relying on tethering Clr4 to a specific chromatin locus. To do this, we generated a reporter system in which *clr4-ΔCD* lacking the chromodomain is fused to Gal4 DNA binding domain (GBD) and tethered to 5 Gal4-binding sites (5xUAS) upstream of *ade6⁺*, inserted at the endogenous *ura4* locus, and with additional intact WT *clr4⁺* available[55]. Artificial tethering of GBD-Clr4-ΔCD resulted in silencing of the *ade6⁺* reporter gene as indicated by red colony formation on low-adenine medium and accumulation of H3K9me2/3 by ChIP-qPCR (Fig. 3e), in accordance with previous studies[55–60]. In contrast, *ade6⁺* silencing was greatly decreased by introduction of *ubc4-1* and *cul4-1* mutations, demonstrating that the ubiquitination activity of Ubc4-CLRC is required for spreading of silencing into the *ade6⁺* reporter (Fig. 3e).

## Phase-separated liquid droplet formation of Clr4 is inhibited by non-coding RNA in vitro

Clr4 and its mouse homolog, mSUV39H1, have an intrinsically disordered region (IDR), also known as a low-complexity domain (LCD), between its chromodomain (CD) and PreSET domain (Supplementary Fig. 8a, b)[51], and IDR/LCD are typically responsible for phase separation[31,33,35,37,38,61]. We found that recombinant GFP-Clr4^Full and GFP-Clr4^IDR (51-220 amino acids of Clr4) proteins readily form liquid droplets in vitro in a salt-dependent manner even without addition of crowding agents (Fig. 4a; Supplementary Fig. 8c)[62]. The formation of

liquid droplets by GFP-Clr4^IDR was reduced by adding 1,6-hexanediol, which dissolves phase-separated liquid droplets by disrupting the hydrophobic interactions of protein-protein or protein-RNA (Supplementary Fig. 8d)[37,63]. Furthermore, photobleaching of inside part of liquid droplets of GFP-Clr4^IDR resulted in slow but steady recovery of fluorescence indicative of dynamic exchange of GFP-Clr4^IDR protein inside of liquid droplets (Fig. 4b and Supplementary Fig. 8e)[31,39,43]. These data indicate an involvement of phase separation for liquid droplet formation of Clr4.

As the oligomerization of HP1 is important to form phase-separated liquid droplets as well as during chromatin compaction, and as SUV39H1 is also able to tether together to form dimers[40,42,64–66], we reasoned that Clr4 may self-associate to form phase-separated liquid droplets. To do this, we mixed recombinant GFP-Clr4 and mCherry-Clr4 and found that these proteins readily associate together to form phase-separated liquid droplets, supporting the idea of Clr4 self-association (Fig. 4c). CTGS during G1/S phase of the cell cycle is accompanied by transcription of ncRNAs and accumulation of sRNAs and the interaction of RNA-binding proteins with RNA are important for LLPS[17,19,31,34]. To investigate the possible role of centromeric ncRNA in Clr4 phase-separated liquid droplet formation, we added 120 nt, synthetic centromeric ncRNA to GFP-Clr4/mCherry-Clr4. Surprisingly, ncRNA inhibited phase-separated liquid droplet formation of Clr4 and Clr4^IDR in a dosage dependent manner (Fig. 4c and Supplementary Fig. 8c), reminiscent of other IDR-containing proteins that interact with RNA[67–69]. We next examined Clr4 self-association via immuno-precipitation using recombinant Flag-Clr4 and Myc-Clr4 proteins in the presence or absence of ncRNA. The weak interaction of Flag-Clr4 with Myc-Clr4 in the absence of ncRNA is consistent with multivalent weak interactions that form phase-separated liquid droplets (Fig. 4d). In contrast, by addition of ncRNA, Clr4 self-association was greatly increased with increasing concentration of ncRNA, indicating that Clr4 self-association induced by ncRNA actually inhibits its phase-separated liquid droplet formation (Fig. 4c, d). Furthermore, mono-ubiquitinated Clr4 did not bind to the centromeric ncRNA compared to unmodified Clr4 protein from ncRNA binding assays using 3xFlag-Clr4 purified from *S. pombe* cells (Supplementary Fig. 8f). Thus, the RNA binding activity of Clr4 is key for its self-association and phase-separated liquid droplet formation, and is inhibited by mono-ubiquitination.

Previously, ncRNA has been shown to inhibit Clr4 enzyme activity[51], but as nucleosomes were used as substrate, this was thought to be due to competition with nucleosome binding. To characterize the effect of ncRNA-mediated self-association on Clr4 H3K9 methylation activity, we fused the N terminal tail of *S. pombe* H3 (A1-K36) with GST (H3N-GST) and used it as a substrate of Clr4 in the absence or presence of ncRNA[70]. Interestingly, Clr4 enzymatic activity for H3K9me3 decreased greatly as the concentration of ncRNA increases, while Clr4 enzymatic activity for H3K9me2 did not change with increasing ncRNA and self-association of Clr4 (Fig. 4d, e). These results indicate that ncRNA binding and Clr4 self-association has an inhibitory effect on Clr4 enzymatic activity for H3K9me3, independent of nucleosome binding.

SUV39H1 forms a complex with HP1, and SUV39H1 and HP1 together colocalize to distinct heterochromatic subnuclear domains (chromocenters) in human cells[43,66,71,72]. HP1α readily forms phase-separated liquid droplets without addition of DNA or nucleosomes in vitro, but HP1β only forms liquid droplets with addition of the N-terminal chromo domain (CD) of SUV39H1 and nucleosomes modified on histone H3K9me3[43]. Like other HP1 proteins, *S. pombe* Swi6 has been shown to undergo liquid-liquid phase separation (LLPS) which contributes to the highly dynamic features of heterochromatin[40,42,73]. Swi6 forms phase-separated liquid droplets only in combination with nucleosomes[73]. As in human cells, we found that recombinant mCherry-Swi6 could not form phase-separated liquid droplets in vitro even in combination with full-length GFP-Clr4. However, GFP-Clr4^IDR

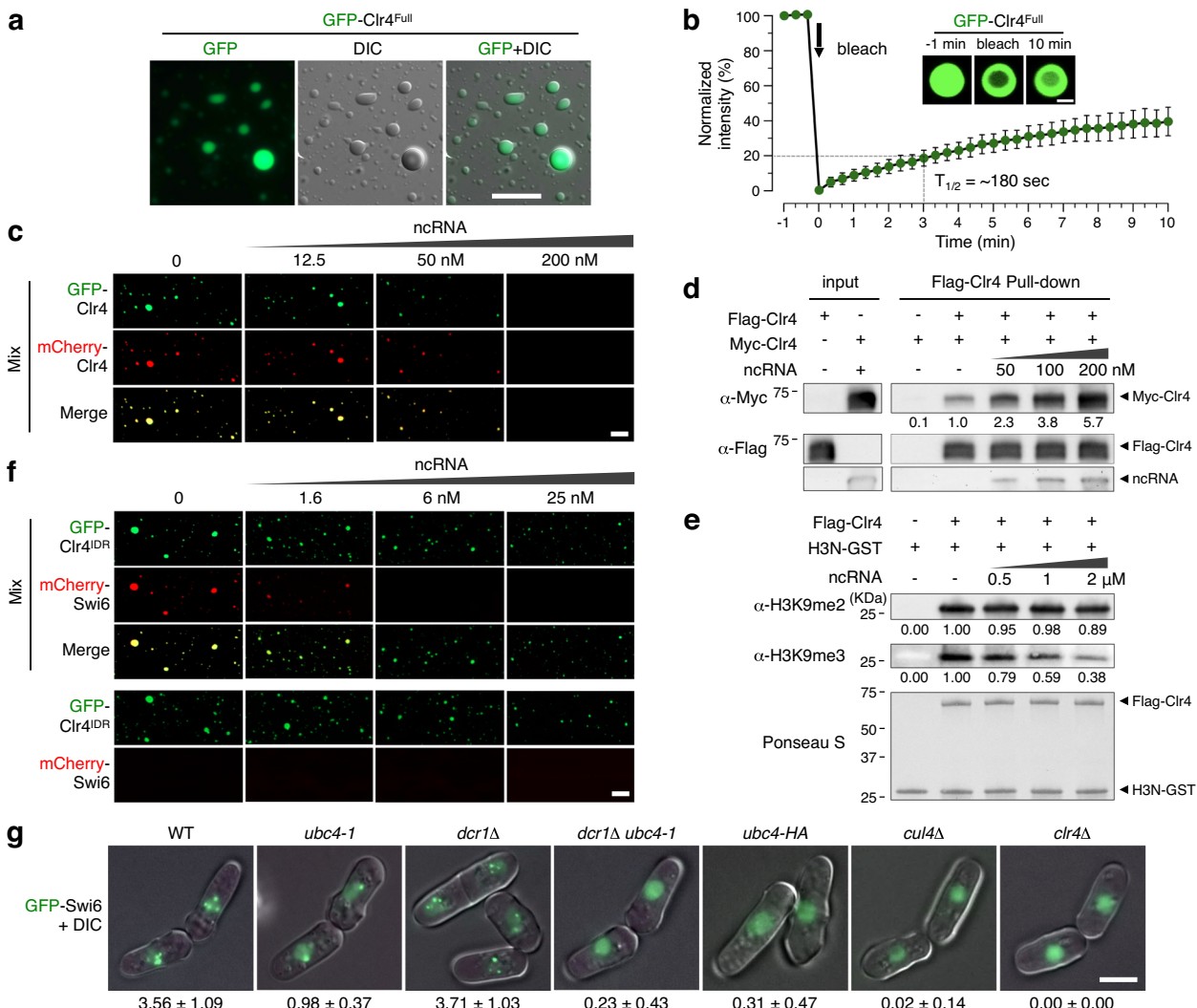

**Fig. 4 | Phase-separated liquid droplet formation of Clr4 is inhibited by non-coding RNA. a** Representative images of phase-separated liquid droplets of GFP-Clr4[Full] protein. DIC; Differential interference contrast. Scale bar, 5 µm. **b** FRAP (Fluorescence Recovery After Photobleaching) of GFP-Clr4[Full] before and after bleaching. The intensity of fluorescence is indicated by mean and S.D. ($n = 5$). An arrow shows the time of bleaching ( ~ 0 min) and halftime of recovery (T$_{1/2}$) is shown. Representative images at times before (−1 min), at (0 min) and after bleaching (10 min) are shown. **c** Representative images of phase-separated liquid droplets of mixed GFP-Clr4 and mCherry-Clr4 (500 nM of each), in the presence of different amounts of centromeric ncRNA. Scale bar, 5 µm. **d** In vitro pull down assay for Flag-Clr4 with Myc-Clr4 (500 nM of each) in the presence of increasing amounts of centromeric ncRNA. WB analyses of input and precipitated proteins are shown. The ratio of Myc-Clr4 precipitated by Flag-Clr4 in the absence of ncRNA was set to 1. ncRNA was stained with SYBR Gold. Molecular weight markers (KDa) are shown and uncropped images are provided in a Source Data file. **e** In vitro histone methylation (HMT) assay. Recombinant H3N-GST (10 µM) was used in HMT assay with Flag-Clr4 (5 µM) in the absence and presence of ncRNA. WB analyses were carried out using H3K9me2 and H3K9me3 antibodies. Ponseau S staining of the blot is shown below. Molecular weight markers are shown and uncropped images are provided in Source Data. **f** Representative images of phase-separated liquid droplets of GFP-Clr4[IDR] mixed with mCherry-Swi6 and unmixed GFP-Clr4[IDR] and mCherry-Swi6 (bottom, 500 nM of each), in the presence of different amount of centromeric ncRNA. Scale bar, 5 µm. **g** Live cell imaging showing condensed GFP-Swi6 heterochromatic foci in indicated strains. Condensates are lost in strong alleles of *dcr1Δ ubc4−1, ubc4-HA, cul4Δ* as well as *clr4Δ* mutants. Scale bar, 2.5 µm. The average number of GFP-Swi6 condensates are indicated below each panel. S.D. ($n > 50$).

and mCherry-Swi6 could form phase-separated liquid droplets when mixed together (Fig. 4f). These droplets dissolved upon the addition of a much smaller amount of ncRNA compared to that required to dissolve droplets formed by mixing GFP-Clr4 with mCherry-Clr4 (Fig. 4c, f). Thus, these results suggest the possibility that phase-separated liquid droplet formation of Swi6 can be promoted by Clr4[IDR] independently of modified nucleosomes, but only during transcriptional silencing when RNA is absent. GFP-Swi6 localizes to heterochromatin in vivo in a H3K9me-dependent manner and we used this to assess heterochromatin condensation in *ubc4* mutant backgrounds (Fig. 4g). Compared to WT and *dcr1Δ* strains which have up to 4 condensed foci representing the centromeric chromocenter, telomere clusters and the mating type locus, *ubc4-1* mutants retained only one

visible chromocenter, presumably due to the complete loss of H3K9me2/3 and Swi6 at telomeres and mating-type loci detected by ChIP (Supplementary Fig. 2c, d). The remaining GFP-Swi6 foci were abolished in *dcr1Δ ubc4*-1, *ubc4-3xHA*, *cul4Δ* and *clr4Δ* mutants in accordance with complete loss of H3K9me2/3 from all heterochromatin in these mutants (Fig. 4g). When a strain which has both GFP-Clr4 and mCherry-Swi6 was used, GFP-Clr4 and mCherry-Swi6 co-localized to a condensed focus in vivo in WT cells. Only one chromocenter was visible because the N-terminal GFP fusion to Clr4 causes a silencing defect (Supplementary Fig. 9a, b). GFP-Clr4 and mCherry-Swi6 co-localization was significantly reduced in *ubc4-1* and *dcr1Δ* and completely abolished in *dcr1Δ ubc4-1* double mutants (Supplementary Fig. 9a). These results indicate that Clr4 and Swi6 form condensates

in vitro and in vivo, which is promoted in the absence of RNA during the G2 phase of the cell cycle. After mitosis, phosphorylation of H3S10 evicts Swi6 from heterochromatin, resulting in transient transcription in G1 and S phase[19]. RNA promotes self-association of Clr4 and rapid dissolution of heterochromatic condensates, consistent with previously observed de-clustering of GFP-Swi6 chromocenters during the cell cycle[74]. Subsequent mono-ubiquitination by CLRC releases Clr4 from RNA and promotes phase-separated liquid droplet formation together with H3K9me3 and Swi6 to drive the transition to TGS in G2.

### Ubiquitination and de-ubiquitination of Epe1 and Bdf2[BRD4] regulates centromeric sRNA and Clr4 for heterochromatin silencing

It has recently been shown that Ubc4 and Cul4 are strongly enriched in Swi6-associated chromatin, along with other CLRC subunits, as well as in chromatin associated with the BET family double bromodomain protein Bdf2 (a homolog of mammalian BRD4)[49]. Bdf2 is recruited by the putative H3K9 demethylase, Epe1, which localizes to centromeric heterochromatin in a Swi6-dependent manner and contributes to sRNA production[75–77], which also occurs mainly in S phase[19]. Bdf2 binds to RNA Pol II promoters at the centromeric heterochromatin boundary[78] and interacts with the TFIID transcription factor complex[79,80] while Epe1 is confined to these boundaries by Cul4-Ddb1[Cdt2] E3 ligase-dependent poly-ubiquitination[75]. We found that the increased centromeric sRNA production in *ubc4-1* and *cul4-1* mutants was accompanied by spreading of Epe1 and Bdf2 to pericentromeric repeats, as well as Ago1 and Chp1, two components of the RITS siRNA complex (Figs. 2f, 5a, b; Supplementary Fig. 10a–c). Moreover, *epe1Δ* and *bdf2Δ* deletion mutants suppressed the increased sRNA production in *ubc4-1* and *cul4-1* mutants back to wild-type levels (Fig. 5a), by reducing recruitment of RNA Pol II (Supplementary Fig. 10d). Consistent with the regulation of Epe1 by Cul4[75], we found that Ubc4 and CLRC induced poly-ubiquitination and degradation of Bdf2 (Fig. 5c and Supplementary Fig. 10e, f). Bdf2[BRD4] recruits TFIID and RNA Pol II to centromeric heterochromatin, an important step in precursor ncRNA transcription for sRNA production[79,80]. To mimic the effect of *ubc4-1* and *cul4-1* on sRNA production, we overexpressed Epe1 (*epe1-OE*, Supplementary Fig. 10g). As expected, *epe1-OE* resulted in increased sRNA production by increased recruitment of Epe1 and Bdf2, but unlike *ubc4-1* and *cul4-1*, it also resulted in decrease of both H3K9me2 and H3K9me3 at centromeric heterochromatin (Fig. 5a, b; Supplementary Fig. 10h–j). Thus, Ubc4-CLRC regulates sRNA production separately through Epe1 and Bdf2, which in combination with Clr4, regulates the proper transition from CTGS to TGS (Supplementary Fig. 10k).

Ubiquitination is a reversible, highly dynamic and transient modification, and its removal is catalyzed by deubiquitinating enzymes[21,81,82]. A deletion mutant of the deubiquitinating enzyme Ubp3 rescues the heterochromatic silencing defects of *ago1Δ*, and *ubp3Δ* shows positive epistatic genetic interactions with CLRC component mutants[83,84]. While deletion of Ubp3 did not increase Clr4 mono-ubiquitination most likely because of redundancy between deubiquitinating enzymes (Supplementary Fig. 11a)[85], the overexpression of Ubp3 (*ubp3-OE*) resulted in a silencing defect at the centromeric *ura4+* reporter gene and reduced mono-ubiquitination of Clr4, supporting the idea that Ubp3 counteracts Clr4 mono-ubiquitination by Ubc4-CLRC (Fig. 5d and Supplementary Fig. 11b, c). The overexpression of Ubp3 also resulted in an increase of H3K9me2 and decrease of H3K9me3, as well as increased sRNA production, with increased binding of Epe1 and Bdf2 to centromeric heterochromatin, resembling the phenotypes of *ubc4-1* or *cul4-1* mutants at centromeric heterochromatin (Fig. 5e, left; Supplementary Fig. 11d). Similarly, at telomeres, *ubp3-OE* resulted in increased H3K9me2 but decreased H3K9me3 as well as greatly increased recruitment of Epe1 and Bdf2, resulting in ectopic sRNA production (Fig. 5e, right; Supplementary Fig. 11e). The 5′-nucleotide of these sRNAs from centromere and subtelomere repeats had a strong U/A bias and the typical sRNA size was 23±1 nt, as is expected for bona fide RNAi products (Fig. 5f, g)[12,86]. These results indicate that Ubp3 counteracts the activities of Ubc4-CLRC, both in terms of H3K9me2/3 transitions and sRNA production, with overexpression of Ubp3, making the chromatin structure of telomeres resemble that of the centromere.

## Discussion

Our findings indicate that, while CTGS and H3K9 di-methylation depend strictly on RNAi, mono-ubiquitination of Clr4 promotes the transition from CTGS to TGS by first releasing Clr4 from heterochromatin. The key lysine residues in the IDR form a β-sheet on binding of the nucleosome core[51], which is likely disrupted by ubiquitination. Deubiquitination by Ubp3 then allows re-engagement and spreading of H3K9 tri-methylation to adjacent chromatin, as demonstrated by ectopic heterochromatin formation at the *ade6+* reporter gene mediated by tethered Clr4 (Fig. 3e). Spreading plays a similar role at the silent mating type locus and at telomeres where heterochromatic silencing is completely lost in *ubc4-1* and *cul4-1* mutants (Fig. 1c; Supplementary Fig. 2c, d). The ability to induce the spreading of silencing by this mechanism may be critical at these genomic loci as these chromosomal locations have only limited capacity for ncRNA transcription and RNAi.

Ubc4-CLRC likely has additional functions, such as H3K14 ubiquitination, which was recently shown to enhance H3K9 methylation in vitro[70]. But H3K9me2 and H3K9me3 are both lost at centromeres in H3K14R mutants, which suggests that H3K14 ubiquitination is essential for histone H3K9 methylation but not for transition of H3K9me2 to H3K9me3 (Supplementary Fig. 12a–c). Centromeric sRNA production was not affected by mutants in the chromo- (*clr4-W31G*), and SET domains (*-F449Y* and *-I418P*) which are also defective in the H3K9me2-to-H3K9me3 transition (Supplementary Fig. 12d, e)[18]. Interestingly, we found that *clr4-W31G*, *-F449Y* and *-I418P* mutants also have reduced Clr4 mono-ubiquitination indicating a strong correlation between Clr4 mono-ubiquitination and transition of H3K9me2 to me3 (Supplementary Fig. 12f, g). Recently, phosphorylation of the Clr4 SET domain by Cdk1 was also shown to be required for H3K9me3 during meiosis, and for proper gametogenesis, although the role in silencing, if any, was not examined[87]. The hierarchical relationship between phosphorylation, auto-methylation and mono-ubiquitination of Clr4 remains to be explored[52].

Like Swi6, Clr4 forms biomolecular condensates in vitro and this property is mediated by the IDR. Post-translational modifications of IDRs, and their association with RNA, can affect protein function significantly; including the induction of disorder-to-order transitions, alternative complex formation and differential subcellular localization[88–94]. We found that ubiquitination of the IDR of Clr4 prevents RNA binding and protein self-association. This promotes phase-separated liquid droplet formation, and is critical for the CTGS to TGS transition in vivo. These phase-separated liquid droplets of Clr4 and Clr4[IDR]/Swi6 are rapidly dissolved in the presence of ncRNA and this RNA-dependent interaction promotes Clr4 self-association instead (Fig. 4c, d, f; Supplementary Fig. 8c). Intriguingly, binding of centromeric ncRNA to the Swi6 hinge region, which is also disordered and lysine-rich, has previously been shown to prevent Swi6 binding to chromatin[95], consistent with our results. In the absence of mono-ubiquitination, Clr4 recruitment to genomic loci where ncRNA and sRNA are overproduced greatly increased in agreement with previous studies that mammalian SUV39H1 chromodomain binds to pericentromeric RNA for the recruitment of SUV39H1 to heterochromatin, H3K9me3 deposition and heterochromatin assembly (Fig. 3a–c)[96–98]. We found that ncRNA binding inhibits Clr4 enzymatic activity for H3K9me3 likely through binding of ncRNA to the chromodomain and IDR of Clr4 and subsequent self-association (Fig. 4d, e and Supplementary Fig. 8c)[54].

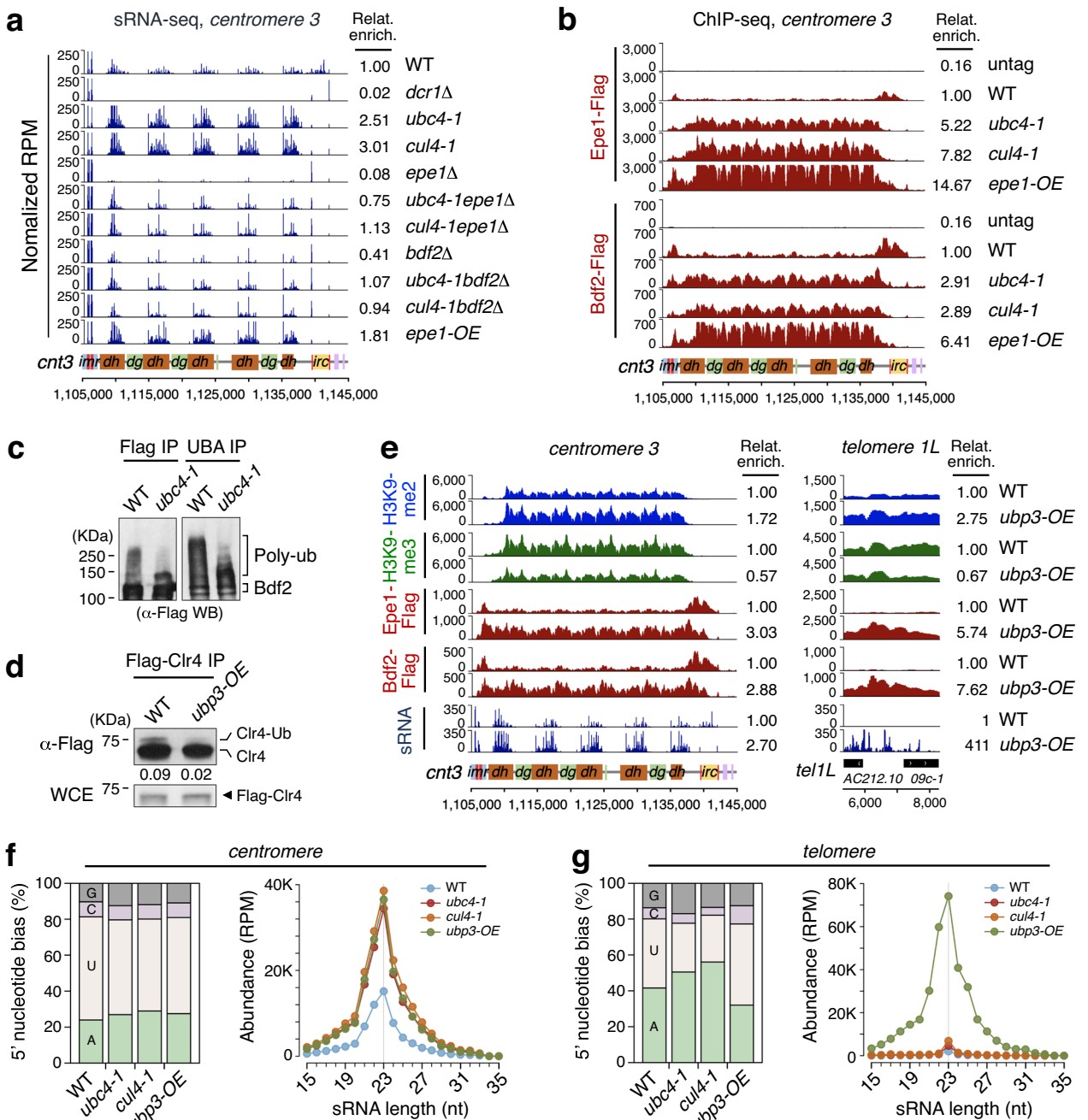

**Fig. 5 | Ubiquitination and de-ubiquitination of Epe1 and Bdf2^BRD4 regulates centromeric sRNA and Clr4 for heterochromatin silencing. a** sRNA-seq reads were mapped to centromere 3 right arm. sRNA peaks remaining in *dcr1Δ* are tRNA-related products. Total counts of sRNA reads without tRNA from different strains were divided by total count in WT (Relative enrichment). *epe1-OE*, *epe1* over-expression strain. Epe1 is a putative H3K9me3 demethylase. **b** ChIP-seq reads of Epe1-Flag and Bdf2-Flag mapped to chromosome 3 centromere (*cnt3*) right arm in the indicated strains. Bdf2 is the homolog of the human co-transcriptional activator BRD4. **c** WB analyses of immunoprecipitated Bdf2-3xFlag (Flag IP) from WT and *ubc4-1* mutant cells treated with proteasome inhibitor MG-132 (100 μM) for 4 h before harvest (left). Bdf2-3xFlag enrichment by ubiquitination affinity beads (UBA IP) are shown on right. Molecular weight markers are shown and uncropped images are provided in Source Data. **d** WB analyses of immunoprecipitated Flag-Clr4 (Flag-Clr4 IP) in the indicated strains. *Ubp3-OE*, de-ubiquitinase *ubp3* overexpression strain. Molecular weight markers are shown and uncropped images are provided in Source Data. **e** ChIP-seq of H3K9me2, H3K9me3, Epe1-Flag and Bdf2-flag and sRNA-seq reads at centromere 3 (left) and telomere 1L (right) in the indicated strains. **f, g** 5′-nucleotide composition (left) and size distribution (right) of centromeric sRNAs (**f**) and telomeric sRNAs (**g**) in the indicated strains.

In summary (Fig. 6), we have found that ncRNA transcription and sRNA production during G1/S promote CTGS and H3K9 di-methylation through recruitment of CLRC. The production of sRNA depends on the transcriptional co-activator Epe1-Bdf2, which licenses recruitment of RNA Pol II to transcriptionally permissive heterochromatin[13–15,18,46,53,59,99,100]. Subsequently, removal of Epe1-Bdf2 by Ubc4-CLRC-mediated poly-ubiquitination silences transcription, while Clr4 mono-ubiquitination is required for its dissociation from chromatin, and from ncRNA, triggering Clr4 dependent H3K9me3 and subsequent binding of HP1 proteins, Swi6 and Chp2 important for RNAi-independent TGS[18,53,101]. In the absence of RNAi, variable amounts of H3K9me2 are maintained at the centromere, indicating that Clr4 must be recruited, at least in part independently of RNAi[102]. Based on our results, one explanation is that ncRNA, in the presence of existing

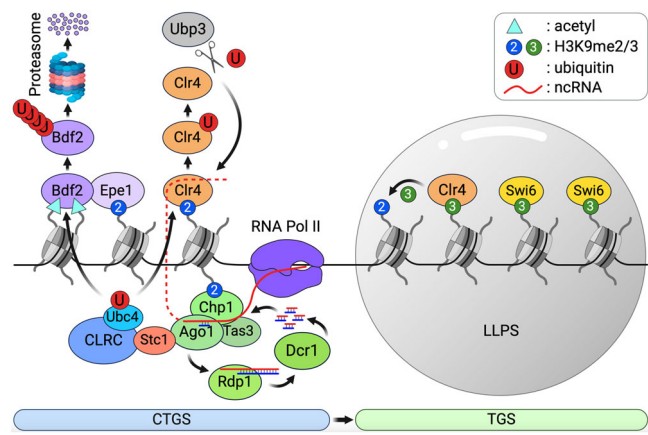

**Fig. 6 | Model for heterochromatin phase transition.** The histone H3 lysine-9 (H3K9) methyltransferase Clr4[SUV39H1] is recruited to chromatin by interactions with methylated H3K9 and with non-coding RNA, which is processed by RNAi (dotted red line). Ubc4-CLRC is recruited by RNAi, and mono-ubiquitinates Clr4[SUV39H1], releasing it from chromatin and from non-coding RNA. Release promotes phase separation along with Swi6, and the transition from H3K9me2 (blue) to H3K9me3 (green) in heterochromatin. Ubc4-CLRC also poly-ubiquitinates Bdf2[BRD4], a transcriptional co-activator required for Pol II transcription and small RNA production. In this way, Ubc4-CLRC mediates the transition from co-transcriptional gene silencing (CTGS), mediated by RNAi, to transcriptional gene silencing (TGS) mediated by phase separation of H3K9me3 and Swi6[HP1]. Created in BioRender. Kim, H. (2024) BioRender.com/r28w184.

H3K9 methylation, initially recruits Clr4 and promotes self-association and chromatin binding. In S phase, RNAi recruits the Ubc4-Cul4-Rik1[Raf1Raf2] E3 ligase[59], which mono-ubiquitinates Clr4 preventing RNA binding, and promoting H3K9me3, potentially by phase-separated liquid droplet formation and three-dimensional "spreading" along with Swi6 (Fig. 6). Removal of ubiquitin moieties by Ubp3 allows rapid recycling of Epe1-Bdf2 and of Clr4, ready for the next cell cycle. Given the high conservation of Ubc4, Clr4, Brd4 and Cul4, similar transitions between RNAi-dependent gene silencing and heterochromatin assembly may be regulated by ubiquitination and RNA in other organisms, including mammals, where TRIM28, HP1 and SUVH39 form condensates with H3K9me3 nucleosome arrays, similar to those reported here[43].

## Methods

### Yeast strains and plasmids manipulation

Yeast strains, plasmids, oligonucleotide sequences and antibodies are listed in Supplemental Tables S1–S4. All yeast strains were constructed by the PCR-based targeting method[103] and lithium-acetate transformation method[104]. For N-terminus tagging of 3xFlag to *clr4*, *rik1* and *raf1*, endogenous promoters and terminators of each gene were used. For C-terminus tagging of 3xFlag, 3xFlag sequence and *adh1* terminator with Kan resistant (Kan[R]) gene was inserted at the end of open reading frames (ORF). For some 3xFlag-Clr4 lysine mutants, *adh1* terminator with Kan resistant (Kan[R]) gene was inserted at the end of *clr4* open reading frames (ORF). For overexpression of *ubp3*, *ubp8*, and *epe1*, 500 bp of *adh1* promoter was inserted between their promoter and ORF. To construct pET-14b-Clr4, the coding DNA sequence of full-length Clr4 (amino acids 1-490) was amplified from *S. pombe* genomic DNA, digested with *NdeI* and cloned into *NdeI*-digested pET-14b. For pHis-parallel-GFP-Clr4[Full] and pHis-parallel-GFP-Clr4[IDR] construction, the coding DNA sequence of full-length Clr4[Full] (amino acids 1-490) and Clr4[IDR] (amino acids 51-220) were amplified from *S. pombe* genomic DNA, digested with *BamHI/XhoI* and cloned into *BamHI/XhoI*-digested pHis-parallel-GFP. For pHis-parallel-mCherry-Clr4, mCherry-Clr4 DNA sequence was synthesized (IDT), digested with *NdeI/XhoI* and cloned

into *NdeI/XhoI*-digested pHis-parallel-GFP. For pHis-Clr4[IDR]-GFP, DNA sequence was synthesized (IDT) and cloned into pET-14b vector using In-Fusion HD cloning kit (Takara). To construct pET-14b-Flag-Clr4 and pET-14b-Myc-Clr4, the coding DNA sequence of full-length Clr4 (amino acids 1-490) was amplified from *S. pombe* genomic DNA using oligomers containing 6xHis-1xFlag or 6xHis-1xMyc sequences respectively and cloned into pET-14b vector using In-Fusion HD cloning kit (Takara).

### Silencing spot assay

Cells grown freshly on YES plate were picked and resuspended in water to make a concentration of ~5 × 10⁴ cells per µl. 8 µl of 10-fold serial dilutions were spotted on appropriate plates like YES, low adenine (Low Ade), EMM-Ura (-Ura), and YES+TBZ (TBZ), and incubated 3 days at 30 °C for photography. *ura4*⁺ reporter gene silencing was also tested on 5-fluoro-orotic acid (FOA) plates, which kills cells expressing Ura4.

### Iodine staining

Freshly grown yeast cells were spotted on SPA plates to induce sporulation, incubated at room temperature for 3 days, and then stained with iodine vapor for 2–3 min. Staining of starch which is abundant in mature spore produces a dark brown color.

### EMS mutagenesis and whole-genome sequencing for *ubc4-G48D* suppressor

EMS mutagenesis was carried out as previously described in ref. 105. Briefly, freshly grown 1 x 10⁹ yeast cells were washed with distilled water, resuspended in 1.5 ml of 0.1 M sodium phosphate buffer (pH 7.0) and treated with 50 µl EMS (Sigma M0880) for 1 h at 30 °C. 0.2 ml of treated cells was moved to 8 ml of sterile 5% sodium thiosulfate to inactivate EMS. Cells were washed, resuspended in sterile water and plated on YES plate. Single colonies from plates were patched on SPA plates to induce sporulation, incubated at room temperature for 3 days, and then stained with iodine vapor for 2–3 min. For whole-genome sequencing for *ubc4-G48D* suppressors, genomic DNA was purified from freshly grown yeast cells using Genomic-tip 20/G column (QIAGEN 10223). DNA library was constructed using the TruSeq DNA PCR-Free Library Prep Kit (Illumina 20015962). Barcoded DNA library was sequenced using Illumina MiSeq platform, which generates paired-end 151 nt reads and analyzed as previously described in ref. 106. Briefly, reads were adapter-trimmed and quality-filtered using Sickle (paired-end mode), then mapped to the *S. pombe* genome using Bowtie2[107]. Duplicate reads were removed using Samtools (≤ 0.14% per library). SNPs were called using FreeBayes to find SNPs present in suppressor strain but absent in the parental strain[108]. This uncovered a G→C transversion at position chr2:716,192, a second-site mutation in the *ubc4* gene. The mutation was verified by Sanger sequencing and confirmed to be causative by constructing the same fresh mutation in WT and *ubc4-1* strains.

### Live cell imaging

Images of freshly growing cells were collected using Zeiss Axio Imager M2 with DIC and EGFP channels and processed using Zeiss Zen software.

### Quantitative reverse transcription PCR (RT-qPCR)

Total RNAs were purified from freshly grown yeast cells using *Quick*-RNA Fungal/Bacterial Miniprep Kit (Zymoresearch R2014). Genomic DNAs were removed using DNase I (Roche 04716728001) and RNAs were cleaned using RNA Clean & Concentrator (Zymoresearch R1013). Reverse transcription was carried out using SuperScript IV First-Strand Synthesis System (ThermoFisher 18091050) with random hexamer or oligo dT as primers. For quantitative PCR (qPCR), iQ SYBR Green

Supermix (Bio-rad 1708882) was used with the primers listed in Supplemental Table S3 and samples were run on CFX96 Real-Time PCR Detection System (Bio-rad).

## Chromatin Immunoprecipitation (ChIP)

Chromatin Immunoprecipitation (ChIP) experiments were performed as previously described in ref. 79. Briefly, freshly grown 40 ml cells in YES were fixed with 1% formaldehyde from 37% stock (Sigma F8775) for 20 min at room temperature for H3K9me2, H3K9me3, RNA Pol II, Ago1 and Chp1 ChIP. For Swi6 and 3xFlag-Clr4 ChIP, 40 ml cells were pre-incubated at 18 °C for 2 h and fixed with 1.45% Formaldehyde from 16% stock (ThermoFisher 28908) for 30 min at 18 °C. Cells were quenched with 360 mM glycine and 2.4 mM Tris for 5 min. Whole-cell extracts (WCE) were prepared using FastPrep-24 and sonicated using Covaris. 1–2 µg of antibodies like anti-H3K9me2 (Abcam ab1220), anti-H3K9me3 (Absolute Antibody Ab00700-1.26), anti-RNA polymerase II (BioLegend 904001), Swi6 (Abcam ab188276), Ago1 (Abcam Ab18190), Chp1 (Abcam ab18191) and anti-Flag (Sigma F1804) were pre-incubated with 50 µl Protein A and Protein G sepharose (GE 17513801 and 17061801) mix for 3-4 h and then incubated with sheared chromatin overnight at 4 °C. After washing, bead-antibody-chromatin mixture were treated with Proteinase K (ThermoFisher 100005393) for 1 h at 42 °C and incubated 5-6 h at 65 °C to reverse cross-links. DNA was cleaned up using ChIP DNA Clean & Concentrator (Zymo Research D5205) and subsequently used for qPCR or ChIP-Seq.

## ChIP-seq

ChIP-Seq libraries were constructed using NEBNext Ultra II DNA Library Prep Kit for Illumina (NEB E7645) and barcoded multiplex libraries were sequenced using Illumina MiSeq platform, which generates paired-end 101 bp reads. Raw reads were trimmed using Trimmomatic, mapped to *S. pombe* genome using Bowtie2 and visualized in IGV genome browser. Total ChIP-seq read counts for defined region were analyzed using MultiBigwigSummary or MultiBamSummary.

## sRNA-seq

For sRNA-seq, total RNAs were purified from freshly grown yeast cells using *Quick*-RNA Fungal/Bacterial Miniprep Kit (Zymoresearch R2014) and sRNA were enriched using RNA Clean & Concentrator (Zymoresearch R1013). sRNA libraries were constructed using NEBNext Small RNA Library Prep Set for Illumina (NEB E7330) and barcoded multiplex libraries were sequenced using Illumina MiSeq platform, which generates single-end 36 nt reads. Reads were adapter-trimmed using Cutadapt se, keeping a minimum read size of 15 nt, and mapped to the *S. pombe* genome using Bowtie[109] allowing up to 1 mismatch. The strand-specific genomic coverage was calculated using Bedtools[110].

## Co-Immunoprecipitation (Co-IP) and Western blotting (WB)

Whole-cell extracts (WCE) were prepared from freshly grown yeast cells in lysis buffer (40 mM Tris-HCl, pH 7.5, 1 mM EDTA, 10% Glycerol, 0.1% NP-40, 300 mM NaCl) with Protease inhibitor cocktail (Roche 11836170001). For Co-IP of Cul4-GFP with Ubc4, 2 µg of GFP antibody (Roche 11814460001) was pre-incubated with 50 µl Protein A and Protein G Sepharose (GE Healthcare 17513801 and 17061801) mix for 3-4 h and then incubated with WCE overnight at 4 °C. After washing, purified proteins were boiled and loaded on 4–20% Mini-PROTEAN TGX Gel (Bio-rad 456-1093) and Western blotting was performed with anti-GFP (Abcam 290) and anti-mUBE2D3 (Proteintech 11677-1-AP) antibodies to detect Cul4-GFP and Ubc4 using ECL (Thermo Fisher 32209) assay according to the manufacturer's instruction. For Co-IP of Flag-Rik1, Flag-Dos1 or Dos2-Flag with Ubc4, 50 µl of Anti-Flag M2 Affinity Gel (Sigma A2220) was incubated with WCE overnight at 4 °C. For detection of Flag tagged protein and Ubc4, anti-Flag (Sigma F7425) and anti-mUBE2D3 (Proteintech 11677-1-AP) antibodies were used.

## Purification of ubiquitinated Flag-Clr4 and Bdf2-Flag proteins

For purification of ubiquitinated Flag-Clr4, WCE prepared from freshly grown cells in lysis buffer (50 mM Hepes, pH 7.6, 150 mM NaCl, 300 mM KAc, 5 mM MgAc, 10% Glycerol, 20 mM β-GP, 1 mM EDTA, 1 mM EGTA, 0.1% NP-40, 1 mM DTT, 250 U/ml Dnase I (Roche 04716728001), 25-50 µg/ml Rnase A (Sigma R4642), 1 mM PMSF, 0.3 mM PR-619 (Sigma 662141)) with Protease inhibitor cocktail (Roche 11836170001) and De-ubiquitination/sumoylation Inhibitor (Cytoskeleton NEM09BB) was incubated with Anti-Flag M2 Affinity Gel or Ubiquitination Affinity Beads (UAB, Cytoskeleton UBA01- Beads) overnight at 4 °C. After washing, purified proteins were loaded on 4–20% Mini-PROTEAN TGX Gel (Bio-rad 456-1093). For detection of ubiquitinated Clr4 protein, anti-Flag (Sigma F7425) or anti-ubiquitin (Cell Signalling 3936T) antibodies were used. Cells were also treated with proteasome inhibitor 100 µM MG-132 (APExBIO A2585) for 4 h before harvesting to detect poly-ubiquitinated Bdf2-Flag or to see the effect of blocking proteasome pathway by treating MG-132 on mono-ubiquitinated Flag-Clr4.

## CLRC complex purification from *S. pombe*

Protein purifications were performed as described previously[15]. Briefly, Rik1-TAP were purified from 6 to 12 g of cells. Cells were lysed in 1 volume of lysis buffer (50 mM HEPES-KOH, pH 7.6, 300 mM KAc, 10% glycerol, 1 mM EGTA, 1 mM EDTA, 0.1% NP-40, 1 mM DTT, 5 mM MgAc, 1 mM NaF, 20 mM β-GP, 1 mM PMSF, 1 mM Benzamidine and 1 µg/ml of Leupeptin, Aprotinin, Bestatin and Pepstatin). 1 volume lysis buffer was added and the lysate was centrifuged at 16,500 rpm for 25 min at 4 °C in a SA600 rotor. The first affinity purification was performed using IgG Sepharose beads at 4 °C for 2 h. The beads were washed with lysis buffer and TEV-C buffer (10 mM Tris-HCl, pH 8.0, 150 mM KAc, 0.1% NP-40, 0.5 mM EDTA, 1 mM DTT) and bound protein was eluted using a GST-TEV protease. The eluate was diluted in 2 volumes of CAM-B buffer (10 mM Tris-HCL, pH 8.0, 150 mM NaCl, 1 mM MgAc, 1 mM Imidazole, 2 mM CaCl$_2$, 10 mM βME) and the second affinity purification was performed by adding Calmodulin-Sepharose at 4 °C for 1 h. The beads lysate was washed with CAM-B buffer containing 0.1 % NP-40 and bound protein was eluted with CAM-E buffer (10 mM Tris-HCL, pH 8.0, 150 mM NaCl, 1 mM MgAc, 1 mM Imidazole, 10 mM EGTA, 10 mM βME). Purified CLRC complex was added to in vitro ubiquitination assay of 6xHis-Clr4.

## Recombinant protein purification from E. coli, in vitro ubiquitination and nucleosome binding assay

BL21-CodonPlus (DE3)-RIL strains (Agilent Technologies 230245) that express recombinant Clr4 proteins were grown with 0.4 mM IPTG for 20 h at 18 °C. Induced proteins were purified on HisPur Ni-NTA Superflow Agarose (ThermoFisher 25214) and eluted with elution buffer containing 150 mM Imidazole. The recombinant proteins purified from E. coli were further dialyzed against 1x TBS using Slide-A-Lyzer Dialysis Cassettes (ThermoFisher 66003). For in vitro ubiquitination, recombinant 6xHis-Clr4 protein or 3xFlag-Clr4 purified form *S. pombe* cells were mixed with human Ubiquitin-activating Enzyme (UBE1, R&D systems E-305), human UbcH5c/UBE2D3 (R&D systems E2-627), human Ubiquitin (R&D systems U-100H) or human HA-Ubiquitin (R&D systems U-110), Mg-ATP (R&D systems B-20) and 10x E3 Ligase Reaction Buffer (R&D systems B-71) with or without CLRC complex purified from *S. pombe* cells. For nucleosome or ncRNA binding assay, biotin labeled Recombinant Mononucleosomes (H3.1, Active Motif 31467) and Recombinant Mononucleosomes H3K9me3 (H3.2, Active Motif 31555) or biotin labeled ncRNA (synthesized from IDT) were pre-incubated with Streptavidin beads (GE Healthcare 17-5113-01) and incubated in vitro ubiquitinated 6xHis-Clr4 or Flag-Clr4 proteins purified from *S. pombe* cells. Samples were washed, boiled and loaded on 4–20% Mini-PROTEAN TGX Gel (Bio-rad 456-1093) and WB was performed with anti-Flag (Abcam

ab1791), anti-H3K9me3 (Abcam ab8898) and anti-H3 (Abcam ab1791), and using ECL (Thermo Fisher 32209) assay according to the manufacturer's instruction.

## Mass Spectrometry

The gel band sample was reduced and alkylated with *N*-ethylmaleimide (NEM), digested with trypsin for 1.5 h at 37 °C and washed with ammonium bicarbonate/acetonitrile to remove stain, SDS and other reagents. Peptides were extracted from the gel pieces, dried down and re-dissolved in 2.5% acetonitrile, 0.1% formic acid. Each digest was run by nanoLC-MS/MS using a 2 h gradient on a 0.075 mm x 250 mm C18 column feeding into a Q-Exactive HF mass spectrometer. All MS/MS samples were analyzed using Mascot (Matrix Science, London, UK; version 2.6.1). Mascot was set up to search the cRAP_20150130.fasta; custom4_20190215.fasta; SwissProt_2019_02 database (selected for *Schizosaccharomyces pombe*, unknown version, 5269 entries) assuming the digestion enzyme trypsin. Mascot was searched with a fragment ion mass tolerance of 0.060 Da and a parent ion tolerance of 10.0 PPM. Deamidated of asparagine and glutamine, oxidation of methionine, phospho of serine, threonine and tyrosine, GG of lysine, N-ethylmaleimide of cysteine were specified in Mascot as variable modifications. Scaffold (version Scaffold_4.8.9, Proteome Software Inc., Portland, OR) was used to validate MS/MS based peptide and protein identifications. Peptide identifications were accepted if they could be established at greater than 80.0% probability by the Peptide Prophet algorithm[111] with Scaffold delta-mass correction. Protein identifications were accepted if they could be established at greater than 99.0% probability and contained at least 2 identified peptides. Protein probabilities were assigned by the Protein Prophet algorithm[112]. Proteins that contained similar peptides and could not be differentiated based on MS/MS analysis alone were grouped to satisfy the principles of parsimony. Proteins sharing significant peptide evidence were grouped into clusters.

## Liquid-Liquid Phase Separation (LLPS)

LLPS of GFP-Clr4, mCherry-Clr4, GFP-Clr4$^{IDR}$ and Clr4$^{IDR}$-GFP proteins were induced by diluting proteins to 100–500 nM range in 10 mM HEPES buffer (pH 7.6). To observe the effect of ncRNA on Clr4 LLPS, 120nt synthesized centromeric ncRNA (IDT) was added to the Clr4 proteins. To observe the effect of 1,6-Hexanediol (Sigma H11807) on GFP-Clr4$^{IDR}$ LLPS, GFP-Clr4$^{IDR}$ protein was diluted to 10 mM HEPES buffers (pH 7.6) containing 5% Ethanol or 5% Hexanediol. LLPS was imaged on Zeiss Axio Imager M2 with differential interference contrast (DIC) and EGFP channels and processed using Zeiss Zen software.

## Fluorescence Recovery After Photobleaching (FRAP) experiments

FRAP experiments were performed as previously described in ref. 113. Briefly, 500 nM of GFP-Clr4$^{IDR}$ protein were diluted in 3.6% PEG8000. PEG8000 was used to stabilize phase-separated liquid droplets during the experiments. FRAP experiments were carried out using Zeiss LSM710 Confocal Laser Scanning Microscope with 63x objective. All Bleaching experiments were performed with 488 nm argon laser at 100% intensity. Images before and after photobleaching were taken at 5 s interval and analyzed using standard methods.

## In vitro pull down assay

4 µg of Flag antibody (Sigma F1804) was pre-incubated with 50 µl Protein A and Protein G Sepharose (GE Healthcare 17513801 and 17061801) mix in 1 mL lysis buffer (40 mM Tris-HCl, pH 7.5, 1 mM EDTA, 10% Glycerol, 0.1% NP-40, 250 mM NaCl) for 1 h and then incubated with recombinant Flag-Clr4 and Myc-Clr4 proteins (500 nM each) and different amount of ncRNA (0, 50, 100 and 200 nM) in 500 µL lysis buffer at 4 °C for 3 h. After washing, purified proteins were boiled and loaded on 4–20% Mini-PROTEAN TGX Gel (Bio-rad 456-1093) and

Western blotting was performed with anti-Myc (Abcam ab9106), and anti-Flag (Sigma, F7425) using ECL (Thermo Fisher 32209) assay according to the manufacturer's instruction.

## In vitro histone methylation (HMT) assay

Recombinant Flag-Clr4 protein (5 µM) were mixed with recombinant H3N-GST (10 µM), 200 µM *S*-(5'-Adenosyl)-*L*-methionine chloride dihydrochloride (SAM, Sigma 7007) and different amoount of ncRNA (0, 0.5, 1 and 2 µM) in 50 µL HMT buffer (50mM Tris, pH 8.0, 20 mM KCl, 10 mM MgCl$_2$, 5% Glycerol, 1 mM DTT and 1 mM PMSF) and incubated at 30 °C for 1.25 h. Samples were boiled and loaded on 4–20% Mini-PROTEAN TGX Gel (Bio-rad 456-1093) and Western blotting was performed with anti-H3K9me2 (Abcam ab1220), and anti-H3K9me3 (Absolute Antibody Ab00700-1.26) using ECL (Thermo Fisher 32209) assay according to the manufacturer's instruction.

## Statistics and Reproducibility

The experiments for Figs. 1b, e, 2b, d, 3d, 4a, c–f, 5c–d and Supplementary Figs. 1c, 3, 4, 5e–l, 7c, 8d, f, 10e, f, 11a, 12f, g were performed at least twice and similar results were obtained. The representative results were shown in the figures.

## Source Data File

For an example of presentation of full scan blots, see the Source Data file of https://www.nature.com/articles/s41467-020-16984-1#Sec35 and for more information, please refer to https://www.nature.com/nature-research/editorial-policies/image-integrity.

## Reporting summary

Further information on research design is available in the Nature Portfolio Reporting Summary linked to this article.

## Data availability

Genome-wide datasets are deposited in the Gene Expression Omnibus (GEO) under the accession number GSE156069. Source Data is provided.

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

## Acknowledgements

We thank all Martienssen lab members as well as Danesh Moazed and Mo Motamedi for valuable discussions and suggestions. The authors acknowledge assistance from the Cold Spring Harbor Laboratory Shared Resources, which are funded in part by the Cancer Center Support Grant (5PP30CA045508). This work was supported by the Howard Hughes Medical Institute, and a grant from the National Institute of Health (R35GM144206) (to R.A.M.).

## Author contributions

H.S.K. performed experiments and bioinformatics. B.R. contributed sRNA-seq experiments and bioinformatics. A.Y.C. contributed to EMS mutagenesis, C.H. contributed to LLPS. S.B. LT and AV contributed to protein purification. H.S.K. and R.A.M. designed the experiments and wrote the manuscript.

## Competing interests

The authors declare no competing interests.
