## [Peer Review File · Nature Communications]

Clr4^{SUV39H1} ubiquitination and non-coding RNA mediate transcriptional silencing of heterochromatin via Swi6 phase separationEditorial Note: This manuscript has been previously reviewed at another journal. This document only contains reviewer comments and rebuttal letters for versions considered at *Nature Communications*.

REVIEWER COMMENTS

Reviewer #4 (Remarks to the Author):

The authors have successfully addressed all my concerns. The paper is suitable for publication in its current format.

Reviewer #5 (Remarks to the Author):

In the revised manuscript, the authors addressed some questions, still there are critical issues remain about ubiquitination of Clr4.

1. The main conclusion of this paper is based on the point that Ubc4 can ubiquitinate Clr4. However, the current evidence is not strong to support this point. Most data are based on the slow migration band, which was believed to be ubiquitinated Clr4. If this band can be recognized by anti-Ubiquitin antibody, then the ubiquitinated Clr4 should be detected with anti-ubiquitin antibody in all experiments. I am not satisfied with the authors' response. If this experiment has much more background than the anti-Flag antibody when using anti-Ubiquitin antibody as the author claimed, this band may not be ubiquitination at all. Even if this band disappeared in Ubc4 mutant, it could be something else. As the ubiquitination of Clr4 is critical for this manuscript, they must fix this issue.

2. Since clr4-2-4KtoR mutant lost the ubiquitination, why the authors performed ChIP-seq using clr4-AllKtoR mutant but not clr4-2-4KtoR mutant? The authors claimed that "Because ChIP of clr4-2-8KtoR and clr4-2-4KtoR is slightly stronger than WT but weaker than ubc4-1 or cul4-1, it is not included in this manuscript." Does that mean ubiquitination of Clr4 had negligible effect on Clr4 binding to chromatin? Is it consistent with their conclusion that Clr4 mono-ubiquitination induces its dissociation from chromatin.

3. Fig. 3d: The reason I suggested the authors to add clr4-2-4KtoR mutant in all these in vitro binding assays as a rigor control is that the Clr4 used in the binding assay was ubiquitinated in vitro. It is unknown whether the ubiquitination sites are Clr4 2-4K or something else. If clr4-2-4KtoR mutant is the same with WT unmodified Clr4, then it would be more convincing that the effect is due to Clr4 ubiquitination at 2-4K. Otherwise, the conclusion is too vague.

Reviewer #6 (Remarks to the Author):

The authors have performed additional experiments to address the major questions with respect to phase separation and provided additional support for their model. However, since there are many types of puncta formed within the cell, autophagy, MVB, and the ubiquitin pathway are all accompanied by the formation of puncta. It is not sufficient to judge that phase separation occurs to form membrane-free biomolecular condensates based on the formation of puncta in vivo. Phase separation of proteins in vitro is a very common phenomenon, but it may not occur in cell. The fact that in vivo phase separation occurs requires specific physiological conditions and confirming this conclusion requires very rigorous experiments. The authors need to further confirm the LLPS of Clr4 and Swi6 in vivo.

Major issues:

1. Fig. 3d: First, there is very little difference in the bands of Clr4 before and after the ubiquitination modification in the input, which does not show that "Clr4 was mono-ubiquitinated efficiently by UBE2D3" as stated in line 157 of the article. Second, why is there two bands of ubiquitination of Clr4 detected by HA, but only one band detected by His and Flag antibodies? Finally, the IP results graphs of Fig. 3d in the first version and in this version are different. Why did you change the IP result graph?

2. Fig. 4b: How long is the bleaching time in the FRAP experiment? 1 minute? If the bleaching time is too long, the fluorescence recovery will be low because too much GFP protein is bleached. Besides, the author could not detect droplet fusion, probably because the formed biomolecular condensates are not very fluid. Whether the condensates formed are droplets via LLPS, gel states or solid-like states is a question that needs to be rigorously confirmed. Studies on the gel state and solid-like state can be referred to the following two articles:

<https://doi.org/10.1016/j.cell.2020.11.027>

<https://doi.org/10.1016/j.molp.2023.04.008>

3. Fig. 4d: In previous reports self-association promotes phase separation, in this paper the ncRNA promotes self-association but inhibits phase separation, which is puzzling. How does the negatively charged ncRNA precisely achieve Clr4 dimerization? What is the evidence that Clr4 forms homodimers (Line 257-259)? Fig 4d can only prove that ncRNA promotes Clr4 interactions with itself, not dimer formation.

4. Fig. 4f: Although Wang et al., 2019 in vitro phase separation evidence is insufficient, their in vivo evidence makes up for it. However, in this paper the in vitro evidence used truncated Clr4 and there is no evidence of phase separation in vivo, which is insufficient.

5. Fig. 6: The author states in the response that Clr4 bound to H3K9me3 nucleosome introduces me3 to me2 nucleosome and draws ubiquitinated Clr4 on H3K9me3 in the model. This is against the experimental conclusion in Fig. 3d that H3K9me3 does not bind ubiquitinated Clr4.

6. Fig. 6: What components are referred to in the gray ball in the model that have undergone LLPS? In the author's response mentioned that "we do not demonstrate that ubiquitinated Clr4 is located in the droplet." So why does the droplet contain ubiquitinated Clr4? It is important to note that, instead of liquid-liquid phase separation, chromatin formed a solid-like state in previous reports.

<https://doi.org/10.1016/j.cell.2020.11.027>

7. Supplementary Fig. 9 : The FRAP experiment in vivo are a very crucial part of the conclusions of this paper. Because the formation of puncta in the cell does not only represent the happening of phase separation but may also be the formation of other vesicles with membrane structures. Secondly, the background of the GFP group in Supplementary Fig. 9 is seriously overexposed. If the N-terminal fusion of GFP affects protein expression, maybe try switching to C-terminal. Besides that, why the Swi6's in Fig. 4g have many more puncta in WT, *ubc4*, and *dcr1*, but Swi6 is almost invisible in the Supplementary Fig. 9. Perhaps the authors could try the FRAP experiment in the group of materials in Fig. 4g.

Minor points:

1. Usually, heterochromatic phase transition refers to the change of chromatin state, and S and G2 phase refer to the cell cycle, which cannot be mixed together. Moreover, there is no relevant experiment on chromatin phase separation in the article, but in the model of Fig. 6, it is stated that "Release promotes heterochromatic phase separation along with Swi6...". ... is too absolutely.

2. Line 236-238: The author states that SUV39H1 and SUV39H2 are also able to tether together to form dimers, but the cited articles only indicate that HP1 is able to form dimers.

Rebuttal letter 2- NCOMMS-23-11213-T

We thank Reviewer #4 for acceptance of our manuscript for publication. We have now tried to address remaining questions. All figure numbers in this response refer to the revised manuscript and figures.

Referees' comments:

Reviewer #5 (Remarks to the Author):

In the revised manuscript, the authors addressed some questions, still there are critical issues remain about ubiquitination of Clr4.

1. The main conclusion of this paper is based on the point that Ubc4 can ubiquitinate Clr4. However, the current evidence is not strong to support this point. Most data are based on the slow migration band, which was believed to be ubiquitinated Clr4. If this band can be recognized by anti-Ubiquitin antibody, then the ubiquitinated Clr4 should be detected with anti-ubiquitin antibody in all experiments. I am not satisfied with the authors' response. If this experiment has much more background than the anti-Flag antibody when using anti-Ubiquitin antibody as the author claimed, this band may not be ubiquitination at all. Even if this band disappeared in Ubc4 mutant, it could be something else. As the ubiquitination of Clr4 is critical for this manuscript, they must fix this issue.

→ We present many different experiments to support the fact that Clr4 is mono-ubiquitinated by Ubc4-CLRC complex, such as Flag-Clr4 *in vivo* purification and WB using Flag and ubiquitin antibodies, UBA IP and WB, and Clr4 *in vitro* ubiquitination and Mass-Spec. As the reviewer suggests, disappearance of the Clr4 band by WB using ubiquitin antibody can be something else but this result still clearly supports Clr4 mono-ubiquitination. Importantly, the Clr4 band also disappears in Cul4 and CLRC mutants, strongly supporting ubiquitination rather than other forms of Clr4 modification.

2. Since *clr4-2-4KtoR* mutant lost the ubiquitination, why the authors performed ChIP-seq using *clr4-AllKtoR* mutant but not *clr4-2-4KtoR* mutant? The authors claimed that "Because ChIP of *clr4-2-8KtoR* and *clr4-2-4KtoR* is slightly stronger than WT but weaker than *ubc4-1* or *cul4-1*, it is not included in this manuscript." Does that mean ubiquitination of Clr4 had negligible effect on Clr4 binding to chromatin? Is it consistent with their conclusion that Clr4 mono-ubiquitination induces its dissociation from chromatin.

→ We now include ChIP-seq of *clr4-2-4KtoR* in the revised Fig. 3a. We also re-measured the Flag-Clr4 enrichments and updated them in revised Figures (Fig. 3a). Flag-Clr4 has very strong enrichment precisely where ncRNA and sRNA are highly produced in *ubc4-1* and *cul4-1* mutants and these results support the idea that RNAs might have a specific role in tethering Clr4 to chromatin. *clr4-2-4KtoR* has much weaker binding than *ubc4-1* or *cul4-1* mutants presumably because *clr4-2-4KtoR* mutation does not affect Epe1-Bdf2 pathway which also regulates ncRNA transcription and sRNA

production (Fig. 2f; Supplementary Fig. 6c). We now make this argument in the revised text (Line 193-196).

3. Fig. 3d: The reason I suggested the authors to add *clr4-2-4KtoR* mutant in all these *in vitro* binding assays as a rigor control is that the Clr4 used in the binding assay was ubiquitinated *in vitro*. It is unknown whether the ubiquitination sites are Clr4 2-4K or something else. If *clr4-2-4KtoR* mutant is the same with WT unmodified Clr4, then it would be more convincing that the effect is due to Clr4 ubiquitination at 2-4K. Otherwise, the conclusion is too vague.

→ In the *in vitro* ubiquitination assay, Clr4 can be modified at many lysines in the region which spans the IDR in addition to *clr4-2-4KtoR*, presumably because of the robustness of the *in vitro* ubiquitination assay or because of the unstructured property of the IDR (Supplementary Fig. 5a-d). This is the reason why we didn't try *clr4-2-4KtoR* for *in vitro* ubiquitination and chromatin binding assay.

Reviewer #6 (Remarks to the Author):

The authors have performed additional experiments to address the major questions with respect to phase separation and provided additional support for their model. However, since there are many types of puncta formed within the cell, autophagy, MVB, and the ubiquitin pathway are all accompanied by the formation of puncta. It is not sufficient to judge that phase separation occurs to form membrane-free biomolecular condensates based on the formation of puncta *in vivo*. Phase separation of proteins *in vitro* is a very common phenomenon, but it may not occur in cell. The fact that *in vivo* phase separation occurs requires specific physiological conditions and confirming this conclusion requires very rigorous experiments. The authors need to further confirm the LLPS of Clr4 and Swi6 *in vivo*.

Major issues:

1. Fig. 3d: First, there is very little difference in the bands of Clr4 before and after the ubiquitination modification in the input, which does not show that "Clr4 was mono-ubiquitinated efficiently by UBE2D3" as stated in line 157 of the article. Second, why is there two bands of ubiquitination of Clr4 detected by HA, but only one band detected by His and Flag antibodies? Finally, the IP results graphs of Fig. 3d in the first version and in this version are different. Why did you change the IP result graph?

→ It is much clearer in the Western Blot of input using HA antibody which recognizes only ubiquitinated Clr4. This result shows that Clr4 is ubiquitinated efficiently by UBE2D3. In the *in vitro* ubiquitination assay, Clr4 is modified at several sites within the IDR, presumably because of the robustness of *in vitro* ubiquitination assay or because of the unstructured property of the IDR. We can see 2 bands when we used His antibody, as well as with the Clr4 tag which recognizes both unmodified and ubiquitinated Clr4, likely reflecting mono-ubiquitination of Clr4 in different residues. We added IP result to show that there is no change of Clr4 levels in unmodified and ubiquitinated Clr4 protein (Reviewer #5's comment)

2. Fig. 4b: How long is the bleaching time in the FRAP experiment? 1 minute? If the bleaching time is too long, the fluorescence recovery will be low because too much GFP protein is bleached. Besides, the author could not detect droplet fusion, probably because the formed biomolecular condensates are not very fluid. Whether the condensates formed are droplets via LLPS, gel states or solid-like states is a question that needs to be rigorously confirmed. Studies on the gel state and solid-like state can be referred to the following two articles:

<https://doi.org/10.1016/j.cell.2020.11.027>

<https://doi.org/10.1016/j.molp.2023.04.008>

→ Bleaching time was approximately 10 seconds with 100% strength of laser. Even though it is slow, we could still demonstrate liquid state of Clr4 protein by FRAP assay which is much stronger than chromatin condensates and Arabidopsis Nup62 which show almost no recovery after photobleaching (Strickfaden et al 2020 Cell & Wang et al 2023 Mol Plant). Clr4 is very similar to KMT5C which shows slow but steady recovery after photobleaching (Strickfaden et al 2020 Cell). Clr4 protein, especially Clr4-IDR is

prone to make gel-state when we store protein for a long time at 4°C. We assume that Clr4 has a mid-range of properties between liquid state and solid state.

3. Fig. 4d: In previous reports self-association promotes phase separation, in this paper the ncRNA promotes self-association but inhibits phase separation, which is puzzling. How does the negatively charged ncRNA precisely achieve Clr4 dimerization? What is the evidence that Clr4 forms homodimers (Line 257-259)? Fig 4d can only prove that ncRNA promotes Clr4 interactions with itself, not dimer formation.

→ Transient and multivalent interaction is crucial for LLPS formation. We assume that Clr4 binds ncRNA and Clr4 also self-associates to form dimer based on previous studies of HP1 and SUV39H proteins. It is true, however, that we can't distinguish ncRNA-mediated Clr4 interaction as dimer rather than multimer formation from this assay. We corrected this in the revised text.

4. Fig. 4f: Although Wang et al., 2019 *in vitro* phase separation evidence is insufficient, their *in vivo* evidence makes up for it. However, in this paper the *in vitro* evidence used truncated Clr4 and there is no evidence of phase separation *in vivo*, which is insufficient.

→ Both Clr4 full-length and Clr4-IDR were used for *in vitro* LLPS experiments in our manuscript. Clr4 loses heterochromatin silencing when it has tag at C-terminus and when it has truncation of chromo domain (CD) or SET domain. Clr4-IDR alone without CD and SET domains can't methylate histones, which can dissociate Clr4 and Swi6 proteins like *clr4* deletion mutant (Fig. 4g). This is the reason that we can't do *in vivo* experiment using Clr4-IDR protein.

5. Fig. 6: The author states in the response that Clr4 bound to H3K9me3 nucleosome introduces me3 to me2 nucleosome and draws ubiquitinated Clr4 on H3K9me3 in the model. This is against the experimental conclusion in Fig. 3d that H3K9me3 does not bind ubiquitinated Clr4.

→ By adding ubiquitin to Clr4 in right side of model, we meant to say that after binding of Clr4 to H3K9me3 and its enzymatic activity, Clr4 is ubiquitinated and dissociated from chromatin. In the revised model we have removed ubiquitin and revised Fig. 2g to avoid confusion.

6. Fig. 6: What components are referred to in the gray ball in the model that have undergone LLPS? In the author's response mentioned that "we do not demonstrate that ubiquitinated Clr4 is located in the droplet." So why does the droplet contain ubiquitinated Clr4? It is important to note that, instead of liquid-liquid phase separation, chromatin formed a solid-like state in previous reports.

<https://doi.org/10.1016/j.cell.2020.11.027>

→ The gray ball indicates LLPS of Clr4 with Swi6. As above, we have now removed ubiquitin inside of droplet to avoid confusion. Even though chromatin condensates behave like solid-state but many of chromatin binding proteins or histone modifying proteins behave like liquid-state (Strickfaden et al 2020 Cell).

7. Supplementary Fig. 9 : The FRAP experiment *in vivo* are a very crucial part of the conclusions of this paper. Because the formation of puncta in the cell does not only represent the happening of phase separation but may also be the formation of other vesicles with membrane structures. Secondly, the background of the GFP group in Supplementary Fig. 9 is seriously overexposed. If the N-terminal fusion of GFP affects protein expression, maybe try switching to C-terminal. Besides that, why the Swi6's in Fig. 4g have many more puncta in WT, *ubc4*, and *dcr1*, but Swi6 is almost invisible in the Supplementary Fig. 9. Perhaps the authors could try the FRAP experiment in the group of materials in Fig. 4g.

→ We didn't use the C-terminal GFP tag of Clr4 because it is already known that it hinders *in vivo* function of Clr4. N-terminal tagging is better than C-terminal tagging, but N-terminal tagging of Clr4 also has a silencing defect as shown in Fig. S9b. This is the reason why the GFP-Clr4 mCherry-Swi6 strain has one strong punctum, as silencing of centromeres and other heterochromatic loci is compromised in this strain. Clr4 is an extremely rare protein in the cell and we need very long exposures to see the GFP signal leading to unavoidably increased background, and making *in vivo* FRAP impractical. However, *in vivo* FRAP of Swi6 has been demonstrated (Cheutin et al., 2004 MCB & Stunnenberg et al., 2015 EMOJ) and we show that Clr4 and Swi6 proteins cooperate to undergo LLPS *in vitro*.

Minor points:

1. Usually, heterochromatic phase transition refers to the change of chromatin state, and S and G2 phase refer to the cell cycle, which cannot be mixed together. Moreover, there is no relevant experiment on chromatin phase separation in the article, but in the model of Fig. 6, it is stated that "Release promotes heterochromatic phase separation along with Swi6...". ... is too absolutely.

→ We removed the term "heterochromatic phase" and changed the title and the legend of Fig. 6.

2. Line 236-238: The author states that SUV39H1 and SUV39H2 are also able to tether together to form dimers, but the cited articles only indicate that HP1 is able to form dimers.

→ In the supplementary Fig. S5A of the cited article, they showed that FCCS experiments revealed self-association of soluble SUV39H1 in living cells and the soluble *bona fide* dimeric SUV39H1 protein fraction was calculated to comprise 24 ± 23 % of soluble SUV39H1 protein present at 0.34 μM concentration (Müller-Ott et al., 2014 Mol Syst Biol.).

References

Condensed Chromatin Behaves like a Solid on the Mesoscale In Vitro and in Living Cells.

Strickfaden H, Tolsma TO, Sharma A, Underhill DA, Hansen JC, Hendzel MJ. *Cell*. 2020 Dec 23;183(7):1772-1784.e13. doi: 10.1016/j.cell.2020.11.027. Epub 2020 Dec 15. PMID: 33326747

Phase separation of the nuclear pore complex facilitates selective nuclear transport to regulate plant defense against pathogen and pest invasion.

Wang J, Pei G, Wang Y, Wu D, Liu X, Li G, He J, Zhang X, Shan X, Li P, Xie D. *Mol Plant*. 2023 Jun 5;16(6):1016-1030. doi: 10.1016/j.molp.2023.04.008. Epub 2023 Apr 18. PMID: 37077045

In Vivo Dynamics of Swi6 in Yeast: Evidence for a Stochastic Model of Heterochromatin

Thierry Cheutin, Stanislaw A. Gorski, Karen M. May, Prim B. Singh, Tom Misteli, *Mol Cell Biol*. 2004 Apr; 24(8): 3157–3167. doi: 10.1128/MCB.24.8.3157-3167.2004

H3K9 methylation extends across natural boundaries of heterochromatin in the absence of an HP1 protein.

Stunnenberg R, Kulasegaran-Shylini R, Keller C, Kirschmann MA, Gelman L, Bühler M. *EMBO J*. 2015 Nov 12;34(22):2789-803. doi: 10.15252/embj.201591320. Epub 2015 Oct 5. PMID: 26438724

Specificity, propagation, and memory of pericentric heterochromatin.

Müller-Ott K, Erdel F, Matveeva A, Mallm JP, Rademacher A, Hahn M, Bauer C, Zhang Q, Kaltofen S, Schotta G, Höfer T, Rippe K. *Mol Syst Biol*. 2014 Aug 18;10(8):746. doi: 10.15252/msb.20145377. PMID: 25134515

REVIEWER COMMENTS

Reviewer #5 (Remarks to the Author):

The authors have addressed all my concerns and this manuscript is suitable for publication.

Reviewer #6 (Remarks to the Author):

Review comments uploaded to PDF

Reviewer #6 (Remarks to the Author):

Although the authors have tried to address my questions, I am still very dissatisfied with the work on the Clr4 phase separation part. The authors did not make any experimental effort to confirm the *in vivo* phase separation conclusion, and the newly revised version of the article is very insincere. Unresolved questions have been highlighted under the previous version of the question.

Major issues:

1. Fig. 3d: First, there is very little difference in the bands of Clr4 before and after the ubiquitination modification in the input, which does not show that "Clr4 was mono-ubiquitinated efficiently by UBE2D3" as stated in line 157 of the article. Second, why is there two bands of ubiquitination of Clr4 detected by HA, but only one band detected by His and Flag antibodies? Finally, the IP results graphs of Fig. 3d in the first version and in this version are different. Why did you change the IP result graph?
→ It is much clearer in the Western Blot of input using HA antibody which recognizes only ubiquitinated Clr4. This result shows that Clr4 is ubiquitinated efficiently by UBE2D3. In the *in vitro* ubiquitination assay, Clr4 is modified at several sites within the IDR, presumably because of the robustness of *in vitro* ubiquitination assay or because of the unstructured property of the IDR. We can see 2 bands when we used His antibody, as well as with the Clr4 tag which recognizes both unmodified and ubiquitinated Clr4, likely reflecting mono-ubiquitination of Clr4 in different residues. We added IP result to show that there is no change of Clr4 levels in unmodified and ubiquitinated Clr4 protein (Reviewer #5's comment)

The input of once experiment should strictly correspond to one IP result graph, not one graph for multiple uses. There are ten western graphs in total, and only changing two of them is a data selection bias.

2. Fig. 4f: Although Wang et al., 2019 *in vitro* phase separation evidence is insufficient, their *in vivo* evidence makes up for it. However, in this paper the *in vitro* evidence used truncated Clr4 and there is no evidence of phase separation *in vivo*, which is insufficient.

→ Both Clr4 full-length and Clr4-IDR were used for *in vitro* LLPS experiments in our manuscript. Clr4 loses heterochromatin silencing when it has tag at C-terminus and when it has truncation of chromo domain (CD) or SET domain. Clr4-IDR alone without CD and SET domains can't methylate histones, which can dissociate Clr4 and Swi6 proteins like *clr4* deletion mutant (Fig. 4g). This is the reason that we can't do *in vivo* experiment using Clr4-IDR protein.

(1) In Figure 4f the authors only mix the proteins GFP-Clr4^{IDR} and mCherry-Swi6 to try to prove that the conclusion described in line 287 "phase-separated liquid droplet formation of Swi6 can be promoted by Clr4" is very uncritical. Just using a segment of the IDR region to drive protein phase separation *in vitro* is very likely to be an artefact. Why didn't you do a mix of full-length GFP-Clr4 and mCherry-Swi6 since you

have already obtained the full-length GFP-Clr4 (Fig 4C)? In addition to this, to prove the conclusion of line 287, the droplets after mixing full-length GFP-Clr4 and mCherry-Swi6 also need to perform FRAP.

- (2) Most importantly, experimental evidence of *in vivo* phase separation is essential, at least to demonstrate Clr4 full-length protein phase separation *in vivo* using the FRAP.
- (3) In addition, the 4f figure does not support the conclusions of line 285 to 286. Please check the description of the picture and text carefully for consistency.

281 heterochromatin^{57,59,61}. Swi6 forms phase-separated liquid droplets only in combination
282 with nucleosomes⁶¹. Similarly, we found that recombinant mCherry-Swi6 alone could
283 not form phase-separated liquid droplets *in vitro*. However, GFP-Clr4^{IDR} and mCherry-
284 Swi6 could form phase-separated liquid droplets when mixed together (Fig. 4f). These
285 droplets dissolved upon the addition of a much smaller amount of ncRNA compared to
286 that required to dissolve droplets formed by mixing GFP-Clr4 with mCherry-Clr4 (Fig.
287 4f). Thus, phase-separated liquid droplet formation of Swi6 can be promoted by Clr4
288 independently of modified nucleosomes, but only during transcriptional silencing when
289 RNA is absent. GFP-Swi6 localizes to heterochromatin *in vivo* in a H3K9me-dependent
290 manner and we used this to assess heterochromatin condensation in *ubc4* mutant

3. Supplementary Fig. 9: The FRAP experiment *in vivo* are a very crucial part of the conclusions of this paper. Because the formation of puncta in the cell does not only represent the happening of phase separation but may also be the formation of other vesicles with membrane structures. Secondly, the background of the GFP group in Supplementary Fig. 9 is seriously overexposed. If the N-terminal fusion of GFP affects protein expression, maybe try switching to C-terminal. Besides that, why the Swi6's in Fig. 4g have many more puncta in WT, *ubc4*, and *dcr1*, but Swi6 is almost invisible in the Supplementary Fig. 9. Perhaps the authors could try the FRAP experiment in the group of materials in Fig. 4g.

→ We didn't use the C-terminal GFP tag of Clr4 because it is already known that it hinders *in vivo* function of Clr4. N-terminal tagging is better than C-terminal tagging, but N-terminal tagging of Clr4 also has a silencing defect as shown in Fig. S9b. This is the reason why the GFP-Clr4 mCherry-Swi6 strain has one strong punctum, as silencing of centromeres and other heterochromatic loci is compromised in this strain. Clr4 is an extremely rare protein in the cell and we need very long exposures to see the GFP signal leading to unavoidably increased background, and making *in vivo* FRAP impractical. However, *in vivo* FRAP of Swi6 has been demonstrated (Cheutin et al., 2004 MCB & Stunnenberg et al., 2015 EMOJ) and we show that Clr4 and Swi6 proteins cooperate to undergo LLPS *in vitro*.

Again, *in vivo* evidence is the key to proving protein phase separation. Your *in vitro* evidence of protein phase separation is also insufficient (Question 4).

4. Line 236-238: The author states that SUV39H1 and SUV39H2 are also able to tether together to form dimers, but the cited articles only indicate that HP1 is able to form dimers. → In the supplementary Fig. S5A of the cited article, they showed that FCCS experiments revealed self-association of soluble SUV39H1 in living cells and the soluble *bona fide* dimeric SUV39H1 protein fraction was calculated to comprise 24 ± 23 % of soluble SUV39H1 protein present at $0.34 \mu\text{M}$ concentration (Müller-Ott et al., 2014 Mol Syst Biol.). The original article indicated that it was SUV39H1 that formed the dimer, but the authors continue to state that SUV39H1 and SUV39H2 are also able to tether together to form dimers (line 241-242). Again, please check your full text carefully!

240 As the oligomerization of HP1 is important to form phase-separated liquid droplets
241 as well as during chromatin compaction, and as SUV39H1 and SUV39H2 are also able
242 to tether together to form dimers^{57,59,65-67}, we reasoned that Clr4 may self-associate to
243 form phase-separated liquid droplets. To do this, we mixed recombinant GFP-Clr4 and
244 mCherry-Clr4 and found that these proteins readily associate together to form phase-

Supplementary Fig. S5A

Rebuttal letter 3rd- NCOMMS-23-11213-T

We thank Reviewer #5 for acceptance of our manuscript for publication. We have now tried to address remaining questions.

Reviewer #6 (Remarks to the Author):

Although the authors have tried to address my questions, I am still very dissatisfied with the work on the Clr4 phase separation part. The authors did not make any experimental effort to confirm the *in vivo* phase separation conclusion, and the newly revised version of the article is very insincere. Unresolved questions have been highlighted under the previous version of the question.

1. Fig. 3d: First, there is very little difference in the bands of Clr4 before and after the ubiquitination modification in the input, which does not show that "Clr4 was mono-ubiquitinated efficiently by UBE2D3" as stated in line 157 of the article. Second, why is there two bands of ubiquitination of Clr4 detected by HA, but only one band detected by His and Flag antibodies? Finally, the IP results graphs of Fig. 3d in the first version and in this version are different. Why did you change the IP result graph?

→ It is much clearer in the Western Blot of input using HA antibody which recognizes only ubiquitinated Clr4. This result shows that Clr4 is ubiquitinated efficiently by UBE2D3. In the *in vitro* ubiquitination assay, Clr4 is modified at several sites within the IDR, presumably because of the robustness of *in vitro* ubiquitination assay or because of the unstructured property of the IDR. We can see 2 bands when we used His antibody, as well as with the Clr4 tag which recognize both unmodified and ubiquitinated Clr4, likely reflecting mono-ubiquitination of Clr4 in different residues. We added IP result to show that there is no change of Clr4 levels in unmodified and ubiquitinated Clr4 protein (Reviewer #5's comment).

The input of once experiment should strictly correspond to one IP result graph, not one graph for multiple uses. There are ten western graphs in total, and only changing two of them is a data selection bias.

→ We changed one of the IP result panels because we got same results by repeated experiments. Now, we changed all result panels together to make complete set.

2. Fig. 4f: Although Wang et al., 2019 *in vitro* phase separation evidence is insufficient, their *in vivo* evidence makes up for it. However, in this paper the *in vitro* evidence used truncated Clr4 and there is no evidence of phase separation *in vivo*, which is insufficient.

→ Both Clr4 full-length and Clr4-IDR were used for *in vitro* LLPS experiments in this manuscript. Clr4 loses heterochromatin silencing when it has tag at C-terminus and when it has truncation of chromo domain (CD) or SET domain. Clr4-IDR alone without CD and SET domains can't methylate histones, which can dissociate Clr4 and Swi6 proteins like *clr4* deletion mutant (Fig. 4g). This is the reason that we can't do *in vivo*

experiment using Clr4-IDR protein.

(1) In Figure 4f the authors only mix the proteins GFP-Clr4^{IDR} and mCherry-Swi6 to try to prove that the conclusion described in line 287 “phase-separated liquid droplet formation of Swi6 can be promoted by Clr4” is very uncritical. Just using a segment of the IDR region to drive protein phase separation *in vitro* is very likely to be an artefact. Why didn't you do a mix of full-length GFP-Clr4 and mCherry-Swi6 since you have already obtained the full-length GFP-Clr4 (Fig 4C)? In addition to this, to prove the conclusion of line 287, the droplets after mixing full-length GFP-Clr4 and mCherry-Swi6 also need to perform FRAP.

→ We now mention in text that the mix of mCherry-Swi6 and GFP-Clr4-full length didn't promote mCherry-Swi6 droplet formation (New Line 300-302), similar to HP1 β which requires H3K9me3 nucleosomes to form phase-separated liquid droplets in the presence of N-terminal Chromo domain of SUV39H1 (Wang et al., 2019). However, we show that liquid droplet formation of mCherry-Swi6 is promoted by Clr4^{IDR} (which lacks the chromodomain) in the absence of nucleosomes, suggesting that Swi6 and Clr4 have the capacity for liquid droplet formation, consistent with the co-localization of GFP-Clr4 and mCherry-Swi6 *in vivo* (Fig. S9). We don't fully understand the meaning of suggested FRAP assay for mixed full-length GFP-Clr4 and mCherry-Swi6 because as we mentioned in previous rebuttal letter, the FRAP assay of Swi6 is published and shows very rapid recovery *in vivo*. We fixed Line 287 in the text (New Line 305-308).

(2) Most importantly, experimental evidence of *in vivo* phase separation is essential, at least to demonstrate Clr4 full-length protein phase separation *in vivo* using the FRAP.

→ Because of the very low expression of Clr4 (Fig. S9a), *in vivo* FRAP assay is not practical.

(3) In addition, the 4f figure does not support the conclusions of line 285 to 286. Please check the description of the picture and text carefully for consistency.

→ We were emphasizing the fact that the ncRNA amount which is needed to dissolve liquid droplets of mCherry-Swi6 mixed with GFP-Clr4^{IDR} is much lower (Fig. 4f) than the ncRNA amount for dissolving GFP-Clr4 and mCherry-Clr4 (Fig. 4c). We fixed the citation (Fig. 4c, f) (New Line 305).

3. Supplementary Fig. 9 : The FRAP experiment *in vivo* are a very crucial part of the conclusions of this paper. Because the formation of puncta in the cell does not only represent the happening of phase separation but may also be the formation of other vesicles with membrane structures. Secondly, the background of the GFP group in Supplementary Fig. 9 is seriously overexposed. If the N-terminal fusion of GFP affects protein expression, maybe try switching to C-terminal. Besides that, why the Swi6's in Fig. 4g have many more puncta in WT, *ubc4*, and *dcr1*, but Swi6 is almost invisible in the Supplementary Fig. 9. Perhaps the authors could try the FRAP experiment in the group of materials in Fig. 4g.

→ We didn't use the C-terminal GFP tag of Clr4 because it is already known that it hinders *in vivo* function of Clr4. N-terminal tagging is better than C-terminal tagging, but N-terminal tagging of Clr4 also has a silencing defect as shown in Fig. S9b. This is the reason why the GFP-Clr4 mCherry-Swi6 strain has one strong punctum, as silencing of centromere or other heterochromatic loci is compromised in this strain. Clr4 is an extremely rare protein in the cell and we need very long exposures to see the GFP

signal leading to unavoidably increased background, and making *in vivo* FRAP impractical. However, *in vivo* FRAP of Swi6 has been demonstrated (Cheutin et al., 2004 MCB & Stunnenberg et al., 2015 EMOJ) and we show that Clr4 and Swi6 proteins cooperate to undergo LLPS *in vitro*.

Again, *in vivo* evidence is the key to proving protein phase separation. Your *in vitro* evidence of protein phase separation is also insufficient (Question 4).

→ Because of the very low expression of Clr4 (Fig. S9a), *in vivo* FRAP assay is not practical.

4. Line 236-238: The author states that SUV39H1 and SUV39H2 are also able to tether together to form dimers, but the cited articles only indicate that HP1 is able to form dimers.

☞II In the supplementary Fig. S5A of the cited article, they showed that FCCS experiments revealed self-association of soluble SUV39H1 in living cells and the soluble *bona fide* dimeric SUV39H1 protein fraction was calculated to comprise 24 ± 23 % of soluble SUV39H1 protein present at 0.34 μ M concentration (Müller-Ott et al., 2014 Mol Syst Biol.).

The original article indicated that it was SUV39H1 that formed the dimer, but the authors continue to state that SUV39H1 and SUV39H2 are also able to tether together to form dimers (line 241-242). Again, please check your full text carefully!

→ We fixed it in text (New Line 252-253).

REVIEWERS' COMMENTS

Reviewer #6 (Remarks to the Author):

The authors have addressed most of the issues I raised. Phase separation experiments in vivo are essential, but may be relatively difficult for this study, and the authors have weakened the conclusions of the study by using terms such as "in vitro" and "involvement" in the abstract and results. I can endorse it for publication.